nature
human behaviour
# OPEN
# Socioeconomic roots of academic faculty

Allison C. Morgan [1 ✉], Nicholas LaBerge[1], Daniel B. Larremore [1,2], Mirta Galesic[3], Jennie E. Brand [4] and Aaron Clauset [1,2,3 ✉]

Despite the special role of tenure-track faculty in society, training future researchers and producing scholarship that drives scientific and technological innovation, the sociodemographic characteristics of the professoriate have never been representative of the general population. Here we systematically investigate the indicators of faculty childhood socioeconomic status and consider how they may limit efforts to diversify the professoriate. Combining national-level data on education, income and university rankings with a 2017–2020 survey of 7,204 US-based tenure-track faculty across eight disciplines in STEM, social science and the humanities, we show that faculty are up to 25 times more likely to have a parent with a Ph.D. Moreover, this rate nearly doubles at prestigious universities and is stable across the past 50 years. Our results suggest that the professoriate is, and has remained, accessible disproportionately to the socioeconomically privileged, which is likely to deeply shape their scholarship and their reproduction.

Professors play a unique role in the knowledge economy: they both train the next generation of thinkers and generate new scholarship, which informs national policy and advances scientific discoveries. But the professoriate has never represented the sociodemographic characteristics of the population it serves. While the diversity of the educational pipeline has been extensively studied in terms of race and ethnicity[1–3], and the links between parental income and occupational status, and their children's educational attainment are well documented[4–7], there exist comparatively few systematic studies on the socioeconomic roots of professors or how their socioeconomic origins interact with institutional prestige. Analyses of the socioeconomic backgrounds of faculty will both improve our understanding of the social reproduction of the highest levels of academic attainment and scientific influence, and provide a quantitative basis for studies of how representational diversity influences which and what kind of discoveries are made.

From early childhood through advanced degree attainment, family socioeconomic origins shape educational outcomes[8]. Parental socioeconomic status (SES) impacts children's educational attainment in numerous ways. Parents with higher SES, who tend to be college-educated and occupy high-status occupations, cultivate advantaged cultural and social capital for their children[9]. Parents with high SES also engage in 'concerted cultivation', whereby parents prioritize organized activities that facilitate educational success[10], and spend an increasingly large amount of time on developmental activities that advantage their children's attainment[7]. Families with parents with high SES also have more information about schooling and resourceful social networks than working-class families. Research also suggests that middle- and upper-class students request accommodations and attention in school that ultimately advantage them in their attainment[11]. These networks function through power and privilege across racial, ethnic and social class lines, forming unequal opportunities for academic success. SES and race also impact where families live and the schools that children attend. The dispositions and skills of students with high SES are judged differently by teachers and other school personnel who favour the cultural capital of middle- and upper-class youth[12]. Ultimately, parents, teachers and students behave in ways that reproduce class and racial

inequality in academic achievement[9]. Schools in more and less advantaged neighbourhoods differ markedly in resources available to students[13,14]. These factors all contribute to the substantial effects of family SES on postsecondary schooling.

Research has shown notable socioeconomic differences in not only whether individuals attend and complete college, but also where they attend, with more advantaged students attending more selective institutions. Students completing degrees at highly selective institutions are more likely to come from the top 1% of the US income distribution than from the bottom 50%[15]. Students from more disadvantaged socioeconomic backgrounds who attend college, particularly those who attend selective colleges, also have different experiences on campus that are less conducive to academic success than their higher socioeconomic peers[16,17].

Family SES also influences graduate school applications and admissions[18], as well as students' experiences once accepted[19,20]. In fact, graduate and professional degrees are substantially more heritable than other levels of education. Individuals with parents who have a doctorate or professional degree are increasingly overrepresented among doctorate and professional degree holders[20]. Moreover, research on social mobility suggests that the association between parents' SES and their children's status is larger among postgraduate than bachelor's degree recipients[19,20]. Occupational closure plays an important role in the intergenerational transmission of status[21], and an association between children and their parents' occupations (sometimes called micro-class reproduction) has persisted for decades[22]. Unequal social, cultural and financial capital help reproduce micro-class advantages across generations, and the professoriate may be an occupation with a high level of closure and reproduction.

The accumulation and consequences of these tendencies can be expected to limit the socioeconomic diversity of the professoriate. However, the extent and shape of this association, and its precise interactions with gender, race, ethnicity, academic discipline and prestige are less well understood. Past work shows that faculty are increasingly representative of highly educated families[23], while professors from disadvantaged socioeconomic backgrounds report hyperawareness of how those backgrounds impact their careers[24],

[1]Department of Computer Science, University of Colorado, Boulder, CO, USA. [2]BioFrontiers Institute, University of Colorado, Boulder, CO, USA. [3]Santa Fe Institute, Santa Fe, NM, USA. [4]Department of Sociology, University of California, Los Angeles, CA, USA. ✉e-mail: allison.morgan@colorado.edu; aaron.clauset@colorado.edu

**Table 1 | Percentages of faculty by their parents' highest-held degree**

|  | Some HS or less | HS | Some college | College | Masters | Ph.D. |
|---|---|---|---|---|---|---|
| All professors | 5.5 | 13.7 | 9.5 | 19.5 | 29.6 | 22.2 |
| Anthropology professors | 3.1 | 14.9 | 7.3 | 19.4 | 32.1 | 23.1 |
| Biology professors | 6.5 | 14.3 | 11.6 | 19.5 | 26.2 | 21.9 |
| Business professors | 5.6 | 14.5 | 8.4 | 24.1 | 30.9 | 16.6 |
| Computer Science professors | 6.6 | 10.8 | 8.9 | 21.6 | 26.1 | 26.0 |
| History professors | 2.9 | 10.5 | 8.6 | 17.0 | 34.3 | 26.7 |
| Physics/Astronomy professors | 8.2 | 12.1 | 10.2 | 18.3 | 27.3 | 24.1 |
| Psychology professors | 3.7 | 17.4 | 9.9 | 17.1 | 31.1 | 20.8 |
| Sociology professors | 4.5 | 17.4 | 6.9 | 17.0 | 35.3 | 18.8 |
| Survey of Earned Doctorates (NSF) | ← 25.2 → | | 14.0 | 23.1 | 26.0 | 11.8 |
| US population (Census) | 19.2 | 35.6 | 23.1 | 14.6 | 6.5 | 0.9 |

Rates compared with the closest available data on educational attainment of the US adult population when faculty were born and the education levels of the parents of doctoral recipients when faculty started their tenure-track job. Education levels included some high school (HS) or less, HS, some college, college, masters, or Ph.D.

which can make them feel like outsiders in academia[25]. SES may also interact with academic job placement. Vague narrow definitions of merit or 'fit' disadvantage the work of marginalized scholars, particularly racial minorities[26] or those from working-class or poor backgrounds[27], due to differences in social and cultural capital between these applicants and faculty search committees[21]. Structural factors that influence whether and where a person earns a faculty job, as well as difficulties once employed, will influence the composition of the professoriate.

Beyond composition alone, representational diversity is known to improve academic scholarship[28], and a lack of socioeconomic diversity is likely to deeply shape the type of scholarship that faculty produce and the scholars they train. Recent studies have begun quantifying how researcher identity affects research topic choices[29], as well as the evaluation of that work[30,31]. However, it remains generally unclear what scholarship is missing as a result of limited representational diversity across the professoriate, in part because we lack a clear and quantitative view of current diversity and its roots.

Here, we develop such a view by investigating the indicators of childhood SES of US-based tenure-track professors across eight disciplines spanning science, technology, engineering and mathematics (STEM), the social sciences and the humanities. Tenure-track faculty positions are less financially precarious than contingent positions, rendering the socioeconomic representation of the faculty who obtain these privileged positions notable for social reproduction. We consider how SES is systematically related to faculty placement within academia, how the role of SES varies across broad disciplines, and how parental advanced degrees relate to the likelihood that their children become and remain as faculty. We conclude by suggesting how socioeconomic diversity and its historical trends may limit diversity in the professoriate.

## Results

To study the childhood SES of US faculty, we conducted a large survey of tenure-track faculty at Ph.D.-granting departments in the United States from eight academic disciplines. Responses include information on the education levels of parents and the zip code where faculty grew up. We augmented our survey data with national estimates of educational attainment, income and rural or urban classification by zip code from the US Census, National Science Foundation (NSF) Survey of Earned Doctorates (SED), Internal Review Service (IRS), US News and World Report (USNWR) and the National Research Council (NRC), which allowed us to contrast survey characteristics with patterns in the general US population.

We conducted the online survey between summer 2017 and autumn 2020 on a frame of 46,692 current tenure-track faculty across 1,360 Ph.D.-granting departments in Computer Science, Business, History, Psychology, Physics and Astronomy, Sociology, Anthropology and Biology. These eight academic disciplines were chosen for their diversity of scholarship and represent a broad sample of tenure-track faculty at research intensive institutions in the United States. The sample frame was constructed from the online public directories of institutions, allowing us to explicitly compare respondents with the frame.

In total, 7,204 faculty provided information on a parent's level of highest education (15.4% of survey frame) and 4,807 provided the US zip code in which they grew up (10.3%).

**Parents' education.** Faculty tend to come from highly educated families. Nearly a quarter (22.2%) report at least one of their parents holds a Ph.D., and 3.7% of faculty report both parents hold Ph.D.s. Across all eight disciplines, over half (51.8%) of faculty have at least one parent with a masters degree or Ph.D. (29.6% and 22.2%, respectively; Table 1). In comparison, among adults in the United States aligned to when faculty were born, on average, less than 1% held a Ph.D., and just 7.4% held a graduate degree of any kind.

We calculated the relative likelihood that a faculty member has a parent with a Ph.D., compared with either the US adult population or parents of Ph.D. recipients, by estimating upper and lower bounds. These bounds correspond to differences in whether these datasets described individuals (US Census and NSF SED) versus households. Across all eight fields, we estimate that faculty are on average between 12 and 25 times more likely to have a parent with a Ph.D. than the general population, and about twice as likely as other individuals who hold a Ph.D. (Fig. 1 and Supplementary Note C).

The distributions of parents' educational attainment are similar across the disciplines surveyed, suggesting that despite disciplinary differences in scholarship, funding and culture, having a parent with a Ph.D. is universally advantageous for becoming a professor. The rates at which parents of faculty have a college degree or higher have also slightly increased over time, which mirrors broader social trends in the US population (Fig. 2). Women are particularly more likely to have highly educated mothers: 33.3% of women versus 28.3% of men have a mother who holds a graduate degree (two-tailed test, $z = 4.5$, $n = 7,107$, 95% CI = 0.028 to 0.073, $P < 0.001$). Historically, rates of parents' college completion among women faculty are higher than among men faculty; however, this gap has steadily closed over time fully reaching parity for faculty born in 1985 (Fig. 2).

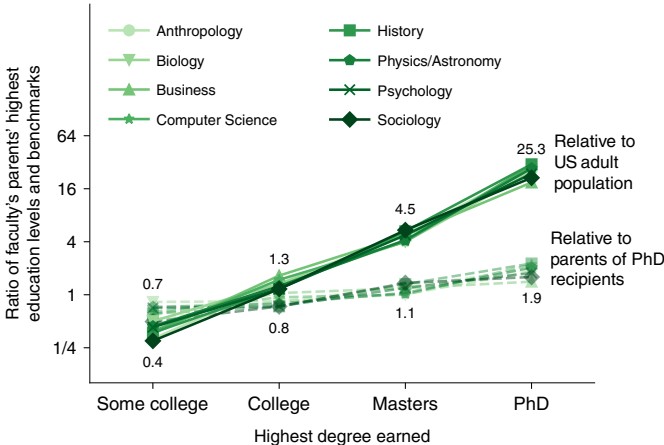

**Fig. 1 | Ratios of faculty parents' education to broader populations.** Percentages of faculty members by their parents' educational attainment levels (n = 7,204) divided by either percentages of the US adult population in those educational attainment levels (solid) or percentages of Ph.D. recipients with parents' in those educational attainment levels (dashed)[68,69]. Disciplines denoted by symbols.

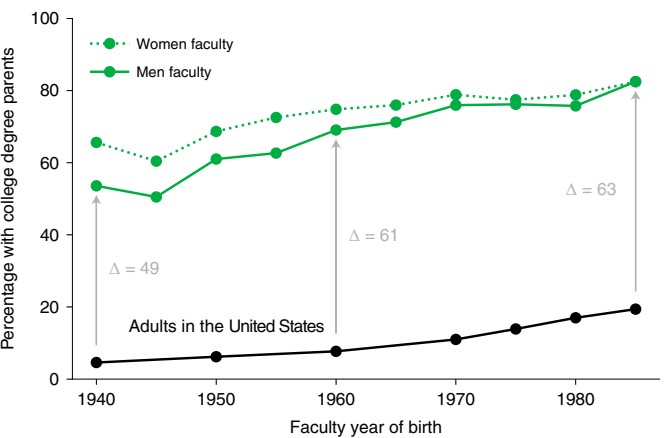

**Fig. 2 | Percentages of faculty with college-educated parents by gender and across time.** Percentages of faculty reporting their parents' highest level of education was at least a college degree by faculty year of birth (green), compared with the fraction of US adults earning a college degree or higher in a given year (black). Grey arrows show the difference between men faculty and US adult trends.

**Academic support.** We find that faculty with Ph.D. parents are far more likely to receive support and encouragement for their academic careers from their parents (on a scale of 1 to 5: 4.5 versus 3.9 for less than Ph.D., two-tailed test, $t = 17.2$, $n = 5,725$, 95% CI = 0.497 to 0.625, $P < 0.001$), which is consistent with prior work[32]. This question came from an optional later section of the survey, but responses were representative along faculty rank and departmental prestige of the frame (Supplementary Note A). In fact, the career support that faculty report receiving from their parents increases with greater parental education (Fig. 3) and does not depend on faculty gender (average rating of 4.0, two-tailed test, $t = -0.2$, $n = 5,695$, 95% CI = −0.072 to 0.057, $P = 0.827$). This strong correlation between parental education and faculty career support suggests that the family resources and experiences of faculty with highly educated parents differ from those without highly educated parents and differ in ways that correlate with improving the odds that a person becomes a professor. For example, the degree of family support is known to correlate with both undergraduate and graduate student retention[19,33]. Given what is known about educational stratification, we expect that there are other sociocultural mechanisms through which being the child of highly educated parents increases the chances of becoming faculty, such as greater identification with academic ideals, more and earlier experiences in activities valued by academia, or simply closer role models[34].

**(Proxied) parents' income.** Faculty also tend to spend their childhoods in wealthier zip codes than do the general public (Fig. 4). The median proxied household income based on zip code data for surveyed faculty when they were children is 23.6% higher than the median across all zip codes (US $73,000 versus US $59,000, Mann–Whitney $U$, $\rho = 0.4$, $n = 1.2 \times 10^8$, $P < 0.001$). Consistent with the importance of parental education on faculty careers, proxied parental income is correlated with parental education: faculty who reported that at least one of their parents holds a college degree were associated with higher average proxied household incomes (US $78,000) than those who said their parents did not hold a college degree (US $59,000; $\rho = 0.3$, $n = 3,916$, $P < 0.001$). Across disciplines, median proxied parental income remains relatively high, ranging from US $67,000 (Sociology) to US $78,000 (History). Faculty are more likely to have grown up in urban areas compared with the geographic distribution of the US population around the

average year faculty were born (89.6% versus 73.6%, point estimates)[35]. And the majority of faculty reported that their parents owned a home during the first 18 years of their life (75.7% versus 13.4% said primarily rented, and 10.9% rented and owned equally, point estimates), higher than one would expect given rates in the United States at the time (62% of homes owned by their occupants in 1960[36]). Hence, faculty tend to come from families with relatively stable childhood financial circumstances.

**Impact of parental education on becoming faculty.** Because the educational attainment of parents is strongly correlated with becoming faculty, we can use data on how many Ph.D.s are granted in a given year to forecast the changing composition of the professoriate. To better quantify this relationship, we model how the likelihood of having Ph.D. parents depends on a scholar's characteristics.

Conditioned on having a parent with a Ph.D., the probability of becoming a faculty member is given by:

$$\Pr(\text{faculty}|\text{Ph.D. Parent}) = \frac{\Pr(\text{Ph.D. Parent}|\text{faculty}) \; \Pr(\text{faculty})}{\Pr(\text{Ph.D. Parent})}$$

where our estimates of $\Pr(\text{Ph.D. Parent}|\text{faculty})$ and $\Pr(\text{Ph.D. Parent})$ are given by the probability of faculty with Ph.D. parents in our survey, and the probability of an adult having a Ph.D. close to the birth year of a professor (22.2% and 0.9%, respectively, Table 1). To estimate the probability of being a tenure-track faculty $\Pr(\text{faculty})$, we assess the proportion of the US adult workforce employed in postsecondary education recorded by the Bureau of Labor Statistics, around the year in which a professor started their job (0.4%)[37]. Because tenure-track faculty are just one kind of employee in postsecondary education, it is likely that this approach overestimates the percentage of the workforce employed as tenure-track faculty.

Combining these quantities, we estimate that the probability of becoming a faculty member given that one's parents hold a Ph.D. $\Pr(\text{faculty} | \text{Ph.D. Parent})$ is 9.5%, indicating a strong degree of both educational heritability and substantial professional advantage.

**Relationships with sociodemographic diversity.** To the extent that becoming a professor is strongly influenced by having parents with doctoral degrees, our results paint a disheartening picture for

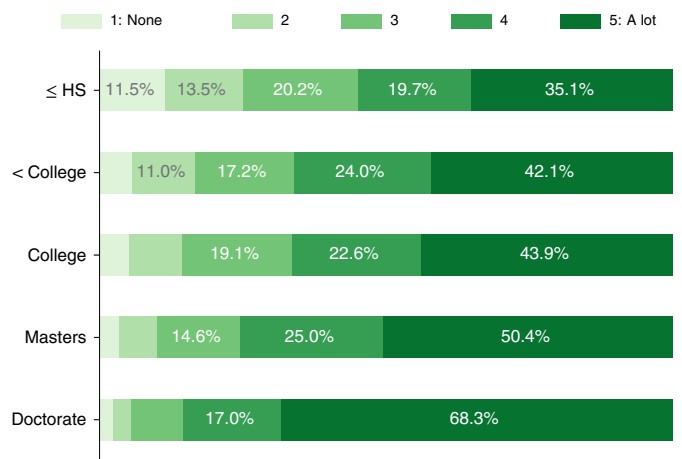

**Fig. 3 | Parental support by education level.** Amount of support parents provided for academic careers on a scale of 1 (None at all) to 5 (A lot), stratified by faculty members' parents' highest education levels.

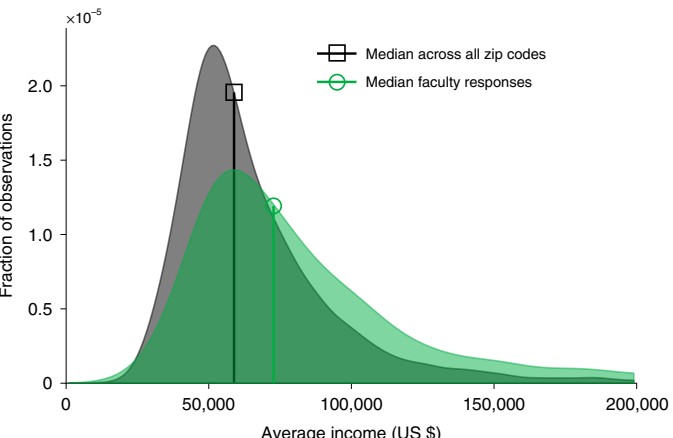

**Fig. 4 | Parental income distribution.** Average income distribution estimated using faculty members' childhood zip codes (green), compared with the income distribution across the 1998 US population (black).

efforts to build a racially diverse pipeline to the professoriate. Broad social and educational inequality within the United States indicates that Black and Hispanic adults are less likely to hold graduate degrees of any kind compared with white adults (Fig. 5). Our race/ethnicity-independent estimate of the probability of becoming faculty may overestimate the production of Black or Hispanic faculty because it fails to account for the lower probability of Ph.D. parents among Black and Hispanic children conditional on SES. Obtaining a precise race/ethnicity-conditioned estimate would require additional conditional estimates that are not currently available (for example, the proportion of Black and Hispanic people who obtain faculty positions).

However, we do find evidence of racial differences within our survey results: white professors are more likely to have a parent with a Ph.D. (23.4%, $n = 5,905$, point estimate) compared with Black or Hispanic faculty (17.2% and 16.9%, respectively, $n = 518$, point estimates). This distinction is even more pronounced among women surveyed, where 25.5% of white women have a Ph.D. parent versus 14.6% of Black women (point estimates, Supplementary Table 3). To the extent that the probability of becoming faculty depends on parental education, and specifically on having Ph.D. parents, this large racial gap in Ph.D. attainment is an intergenerational impediment to the proportion of Black and Hispanic scholars who become tenure-track faculty.

The relationship between parents' education and the gender composition of the professoriate is complex. For instance, women in our sample are more likely to have Ph.D. parents than are men (24.8% versus 20.8%, point estimates). Identifying the reasons underlying this pattern is an interesting direction for future work. Of course, SES does not impede gender diversity in academia in the same way or to the extent that it does for racial diversity. Nevertheless, parents may differentially allocate resources to sons and daughters that influence their academic achievement[38]. Moreover, the intersection between race/ethnicity and gender is not neutral with respect to the likelihood of becoming faculty. For example, Black women faculty are less likely to have Ph.D. parents than are Black men (Supplementary Table 3). Prior evidence suggests that there has been a growing gender gap in college completion, with women outpacing men, among both white and, to a larger extent, Black Americans[39]. The overrepresentation of Black men from educationally privileged families may imply a differential selection within academia, wherein Black men need more social capital to pursue these careers. Untangling how gender,

race/ethnicity and social origins interconnect to shape who pursues academic careers, and specifically why Black women professors were less likely to come from more educationally privileged families, is an important direction for future research.

**Trends in socioeconomic origins.** Academia has undergone many dramatic shifts over the past 100 years, and our survey reveals several interesting and related trends. For instance, we find that the rate of faculty born from 1940 to 1960 reporting that a parent holds a Ph.D. has increased from 13% at the beginning of the period to 26% by the end (Fig. 6a), but then remains stable at above 20% across the next 50 years. This increase from the 1940s to 1960s mirrors the increasing college and graduate school enrolment rates within the United States over the same period[40,41], and hence may simply reflect a general broadening of access to higher education. However, the subsequent stability of the rate at which faculty have a Ph.D. parent, in contrast to the continuing growth in and diversifying demographics of doctoral degree attainment[42], suggests that the relationship between the educational attainment of faculty parents and the likelihood of becoming faculty has been consistent and strong for nearly half a century.

Parental education is so consequential that it also correlates with where in the academic hierarchy a professor lands. Across all years, we find that nearly a third of faculty at top-ranked universities across all eight fields report that one of their parents holds a Ph.D., and faculty at these elite departments are 57.4% more likely to have a parent who holds a Ph.D. than are faculty at the least prestigious departments (29.8% versus 19.0%; two-tailed test, $z = 6.5$, $n = 2,612$ 95% CI = 0.076 to 0.142, $P < 0.001$). This concentration among elite departments is consistent with prior research documenting the ways academic hiring tends to devalue faculty of lower socioeconomic standing [26,27], and advantage faculty from more privileged backgrounds.

**Modelling faculty placement.** Adjusting for faculty discipline, Ph.D. prestige, race/ethnicity and gender within a model, we find that faculty who had parents who attended college tend to be employed at significantly more prestigious universities than faculty without these childhood socioeconomic advantages (Table 2). We recognize that Ph.D. prestige may in fact be endogenous to this model of faculty placement because of Ph.D. placement's likely relationship with parents education, but regardless include it as a control due to the strong evidence of its importance in predicting faculty placement[43]. Here, institutional prestige is defined by

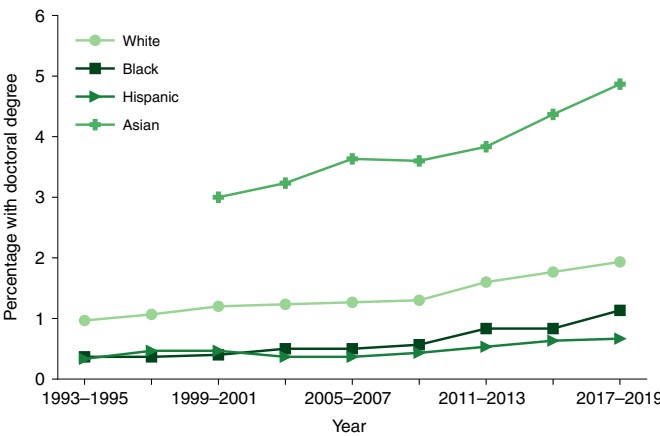

**Fig. 5 | Ph.D. attainment by racial or ethnic group.** Percentages of white, Black, Hispanic and Asian adults (at least 25 years old) in the US population who hold doctorate degrees[84]. Trends are smoothed with averages every three years.

ordinal ranking, where the most prestigious universities have smaller rank (1), and less prestigious have larger rank (100). On average, faculty with a Ph.D. parent move 'up' in the institutional rankings by nearly 7 percentile ranks (two-tailed test, $t = -3.4$, $n = 2,709$, 95% CI $= -10.536$ to $-2.850$, $P = 0.001$). Faculty who grew up in wealthy neighbourhoods, who are also less racially diverse than the general population, also tended to place at more prestigious institutions. SES may constrain an individual's search for a faculty job either by influencing which institutions they apply to, or by shaping their ability to stay on the academic track while searching for employment (Supplementary Note D). These results have direct implications for efforts to increase the socioeconomic and racial diversity of the professoriate, particularly at the most prestigious institutions that train most future professors[43]. The higher-prestige placement of faculty from advantaged socioeconomic backgrounds also represents a structural barrier to the visibility of the ideas of lower SES faculty because scientific discoveries made at more prestigious universities are more likely to spread throughout academia[44].

**Disadvantaged socioeconomic origins.** Despite the significant correlation between having a parent who holds a Ph.D. and placement as faculty in the prestige hierarchy, not all faculty at elite universities (here: top 20% by USNWR or NRC) have this advantage. For elite faculty without Ph.D. parents, career support from colleagues both within and beyond their institution, and the wealth and urbanicity of their childhood zip code all become more important factors in explaining their placement at an elite institution. Faculty without Ph.D. parents who are employed at prestigious institutions report slightly higher levels of support from colleagues compared with those at lower-ranked institutions both within their institution (4.0 versus 3.8; two-tailed test, $t = 3.6$, $n = 4,449$, 95% CI $= 0.078$ to 0.261, $P < 0.001$), and outside their institution (4.0 versus 3.9; two-tailed test, $t = 2.6$, $n = 4,448$, 95% CI $= 0.028$ to 0.204, $P = 0.009$).

Most faculty earned their Ph.D.s at elite institutions[43]. Faculty without a Ph.D. parent, currently employed at an elite university, were more likely to come from more elite Ph.D. programmes than were similar faculty at non-elite universities (5.5 versus 14.0 median; Mann–Whitney $U$, $\rho = 0.3$, $n = 3,813$, $P < 0.001$). Furthermore, without Ph.D. parents, faculty at the top institutions are more likely to come from urban neighbourhoods (92.3% versus 87.3%) that are higher income (US $80,000 versus US $69,000 median; Mann–Whitney $U$, $\rho = 0.4$, $n = 2,974$, $P < 0.001$).

## Discussion

Using a large survey of eight academic disciplines spanning STEM, social science and the humanities, we quantify the extent to which becoming a professor is associated with indicators of SES, and in particular is most accessible to the children of doctoral recipients and those who grew up in wealthy urban neighbourhoods. Considering the intersection between race/ethnicity and class, we also conclude that this dependence on parent educational education and SES is likely to limit racial diversity within the professoriate.

To summarize, nearly a quarter (22.2%) of faculty reported that one of their parents holds a Ph.D., and over half (51.8%) had a parent who holds a graduate degree, compared with less than 10% of US adults of similar ages (Table 1). Faculty who have parents with Ph.D.s report receiving more support from them for their careers (Fig. 3) and are more likely to be employed at elite institutions. Nearly a third of faculty at top-ranked universities report that their parent holds a Ph.D. (29.8%), versus a fifth (19.0%) at lower-ranked institutions. This pattern represents a significant source of social reproduction at the highest levels of academic attainment. Moreover, given broader racial inequality in educational attainment, academia's overrepresentation of inherited advantages represents a fundamental constraint to increasing its racial diversity (Fig. 5).

Previous studies of childhood SES of faculty have often been limited in scope, due to relatively small surveys of faculty or a lack of historical data on indicators of SES in the broader United States, both of which were necessary for this study. Given the well-established correlations between household parental income and child educational attainment and patterns of micro-class reproduction in other economic sectors, the overrepresentation of faculty with Ph.D. parents is perhaps unsurprising. Yet our results quantify just how large, and how persistent, that overrepresentation is.

In fact, the importance of having Ph.D. parents appears so great that the rate of having them nearly doubles across the transition from completing a Ph.D. to obtaining a faculty job (11.8% versus 22.2%; Table 1). This effect indicates a substantial loss of talent in the pipeline from Ph.D. to the transition to a faculty job. Doctoral students with Ph.D. parents may be better prepared for the difficulties of the academic job market, which may confer an advantage that becomes even greater during periods when academic jobs are scarce, for example, during a pandemic or a recession. Furthermore, the stability of this pattern across STEM fields, the social sciences and humanities suggests that the loss of talent at this stage in the pipeline is unlikely to be caused by the existence of attractive non-academic jobs for STEM Ph.D.s. Understanding the causes of this pattern is an important direction for future work.

But the even greater overrepresentation of faculty with Ph.D. parents at the most prestigious institutions (Fig. 6) implies that increasing sociodemographic diversity in academia is not only a pipeline problem. Indeed, our findings complement research that suggests a re-emergence of socioeconomic reproduction at the highest levels of educational attainment[8,45]. Progress towards broadening participation in science will remain limited if our current practices within academia favour individuals with advantages conferred by parental socioeconomic and occupational privilege[21,26,27].

Socioeconomic background plays a critical role in shaping academic success, from secondary school to reaching the ranks of tenure-track faculty positions. Social and cultural factors also play a role[46], such as the cultural value of academic success (which Fig. 3 suggests can involve intergenerational transmission), effective role models or even access to scholarly advice or opportunities from parents who understand how academia works. Ph.D. parents are uniquely positioned to offer guidance and support in children's postsecondary studies, through the graduate application process, negotiating admissions offers, progress in graduate programmes and job market success. For example, within our study, faculty who had a Ph.D. parent also earned their Ph.D.s at slightly more elite

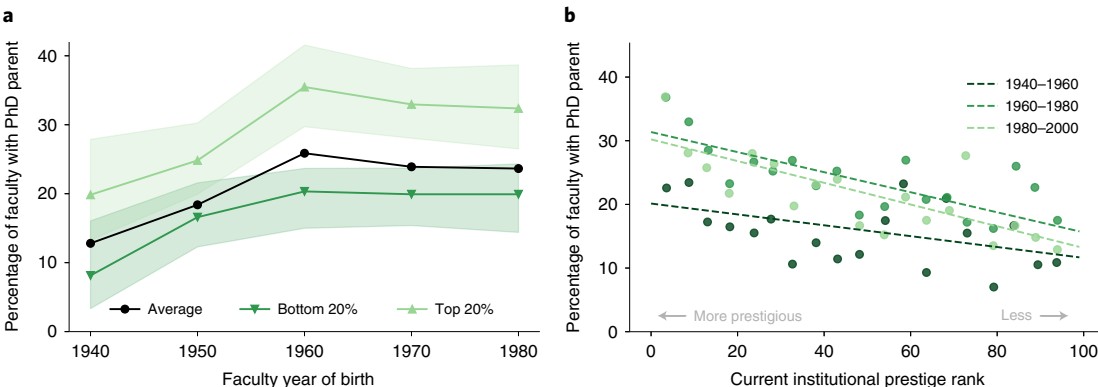

**Fig. 6 | Trends in the percentage of faculty with a Ph.D. parent. a**, Percentage of faculty with at least one parent holding a Ph.D., stratified by prestige of the faculty's current institution. Green upward triangles describe faculty at the top 20% of institutions by USNWR or NRC ranking, and downward triangles the bottom 20% of ranked institutions. Shaded regions denote empirical 95% confidence intervals. The black line describes the average proportion of faculty with Ph.D. parents. **b**, Current institutional ranking of professors surveyed and the percentage of surveyed faculty at that institution who have a parent with a Ph.D. Dots indicate empirical estimates for 1,360 universities of more to less prestige; lines show a stable negative relationship between increasing institutional ranking and having a Ph.D. parent for faculty born in 1940–1960 (dark green), 1960–1980 (medium green) and 1980–2000 (light green).

institutions (7.1 versus 11.6; Mann–Whitney $U$, $\rho = 0.4$, $n = 4,906$, $P < 0.001$), and earning a Ph.D. at an elite institution contributes towards upward academic mobility[47]. This finding suggests that Ph.D. parents' deep understanding of these processes facilitates micro-class reproduction. The overall socioeconomic composition of faculty is likely shaped by several complex, interacting factors, and a full explanation of the patterns will require a careful accounting of each in future work.

The interpretation of our work is limited by the granularity of the IRS and Census data we link to individual responses, and the current composition of tenure-track faculty. Several of our comparisons rely on aggregated measures used to proxy for income based on respondents' childhood zip code. Many zip code boundaries have changed over time and may span heterogenous populations[48]. Each of these may limit the accuracy of our childhood income measures, especially for older faculty. In future work, better estimates of household income could be obtained from information on respondents' childhood Census blocks, or from de-identified historical tax records[49]. Similarly, our analyses of race and ethnicity are limited by the use of the 2010 Census classification standard, which provides some consistency with 2020 standards[50], but precludes certain more fine-grained analyses.

Furthermore, our survey was restricted to current tenure-track faculty at research intensive academic institutions between 2017 and 2020. As a result, our data say little about contingent faculty at these institutions, which represent an increasing fraction of university instructional staff[51]. We also do not know how gender discrimination, racism or classism impacts academics' intentions to persist, and we do not observe faculty who left academia before our survey date, whether that was after graduate school, a postdoctoral position or a contingent or tenure-track academic position, and whether that is due to feeling pulled towards more desirable opportunities outside of academia or pushed out by discrimination in hiring or promotion within. This group may be a population biased toward faculty from lower-income backgrounds[19]. Such a bias would tend to lead us to underestimate the observed associations or importance of SES in our analyses if we aimed to examine associations for all faculty entering academia, but these associations do not bias our estimates for current faculty. An important and complementary direction of future work would examine SES among non-tenure-track faculty and investigate its role in recruitment and retention before and along the tenure track.

While this work highlights the advantages that SES may confer for becoming and remaining faculty, it does not explicitly speak to the social or professional difficulties that underrepresented individuals experience stemming from their gender, race, ethnicity or SES once they become faculty. For example, women are less likely to be viewed as competent[52,53], less likely to be awarded for their research[54], less likely to be invited for talks[55] and less likely to be compensated fairly[56,57]. Black faculty earn less despite no measurable differences in productivity[58] and may have their research evaluated less positively[31], all while facing unequal service burdens and racism on and off campus[59,60]. While research on how SES shapes faculty careers is less comprehensive, work generally points to less supportive environments[24,25] and differences in research or teaching appointments[61]. Individuals with disadvantaged socioeconomic origins may become faculty and still have greater difficulty navigating and finding support within academia.

Future research should consider why the importance of family background varies moderately by gender and across fields. In our survey, women are more likely than men to have a parent with a college degree or higher (Supplementary Table 2), indicating the greater barriers facing women in reaching the highest ranks of academia. The reasons for this difference among tenure-track faculty require more investigation. Percentages of faculty with a Ph.D. parent are lowest among Business and Sociology (17.3%), and highest among Computer Science and History (26.3%, $\chi^2 = 39.1$, $n = 3,362$, $P < 0.001$). Compared with the educational attainment of professors' parents (Table 1), and generally consistent with the result presented here, a survey of lawyers found that about 12% of lawyers were the children of lawyers[62], and among doctors in Sweden, 20% of physicians were the children of physicians[63]. Whether or not having a Ph.D. parent provides more of an advantage to academic faculty than one would expect relative to other high-status occupations is an important question for future research.

Our study's detailed and cross-disciplinary view of the deep socioeconomic roots of current US faculty highlights the social reproduction of the highest levels of academic attainment and scientific influence. These results raise a critical question regarding what ramifications high SES overrepresentation in the academic workforce—and its skewed intersection with race/ethnicity and class in the United States—have on the type of scholarship that faculty produce and on the scholars they train. That is, what discoveries are not made and what ideas are not developed as a result of

**Table 2 | Regression of current institutional prestige based on childhood SES**

| | Model I | Model II | Model III |
|---|---|---|---|
| Urban neighbourhood | 0.129 | 0.552 | 0.914 |
| | [−3.042 to 3.206] | [−2.601 to 3.751] | [−2.255 to 4.361] |
| | $P = 0.938$ | $P = 0.742$ | $P = 0.588$ |
| Avg. income | | −1.447* | −1.352* |
| | | [−4.569 to −0.625] | [−4.381 to −0.578] |
| | | $P = 0.001$ | $P = 0.003$ |
| Parents' highest degree: | | | |
| Elementary | | | −2.946 |
| | | | [−12.995 to 7.747] |
| | | | $P = 0.579$ |
| Some HS | | | −1.816 |
| | | | [−13.870 to 10.393] |
| | | | $P = 0.711$ |
| HS | | | −4.873* |
| | | | [−9.125 to −0.948] |
| | | | $P = 0.024$ |
| College | | | −4.416* |
| | | | [−8.380 to −0.372] |
| | | | $P = 0.030$ |
| Masters | | | −5.177* |
| | | | [−8.743 to −1.570] |
| | | | $P = 0.007$ |
| Ph.D. | | | −6.889* |
| | | | [−10.536 to −2.850] |
| | | | $P = 0.001$ |
| Adjusting for discipline, race/ethnicity, gender, Ph.D. prestige | Yes | Yes | Yes |
| Adjusted $R^2$ | 0.105 | 0.107 | 0.109 |

Institutional prestige is an ordinal ranking from 1 to 100. Covariates include neighbourhood, income and parents' education levels, adjusting for discipline, race/ethnicity, gender and Ph.D. ranking. Bootstrapped 95% confidence intervals in brackets, and coefficients with $P < 0.05$ are denoted by the asterisk. Baseline is women in Anthropology from urban neighbourhoods whose parents' highest education was some college.

academia's historical and current lack of socioeconomic diversity? A lack of diversity is already known to reduce research on public health disparities[31], immigration, mental health and gender-based discrimination[29]. However, it is likely that the loss of useful research is more pervasive, because the relationship between researcher identity and scholarship is complex, depending not only on scientific social capital[64] but also on how that capital can be invested. Developing a quantitative understanding of how representational diversity shapes scholarship across fields is an important direction for future work, which will rely on quantitative assessments of current diversity and its roots like the one provided here. Making academia more diverse and inclusive is a considerable challenge,

given the complex interplay of SES and academic achievement from early schooling through academic appointments. Progress in this direction, however, is likely to produce substantial scientific and societal benefits.

## Methods

Our survey protocol was approved by the University of Colorado Boulder Institutional Review Board. All participants were included in drawings for cash lotteries (Supplementary Note A). Although contingent faculty represent an increasing proportion of the US academic workforce, we surveyed tenure-track faculty because their unique stability in academia makes them more likely to contribute research and mentor scholars over the course of their careers.

Overall, our response rate resembles other online surveys with email invitations conducted in the context of academia[65,66]. While it is not possible to know whether faculty respondents differ in their childhood SES from those who did not respond, respondents were generally representative of their fields by institutional prestige and faculty rank. Additionally, women responded at slightly higher rates than expected in several disciplines (Supplementary Table 1). This pattern may imply a slight upward bias in our analysis of parents' education: in our sample, women are somewhat more likely to come from highly educated families (one of their parents holds a Ph.D.: 24.8% versus 20.8%; $z = 3.9$, $n = 7,145$, $P < 0.001$), across all disciplines (Supplementary Table 2). Roughly a quarter of respondents provided a country outside of the United States where they grew up ($n = 2,007$; 25.1%), consistent with prior estimates of the proportion of international faculty at US universities[67].

We asked faculty to reflect on their childhood and report their parents' highest levels of education during this period (Supplementary Note A). We compared their responses to the adult educational attainment statistics in the US population in the year the survey respondent was born[68], and to educational attainment of parents of US doctoral recipients in the year they graduated from their doctoral programme[69]. These benchmarks, respectively, facilitate comparisons of faculty with the general public, and with academics at a career stage just prior to their current one. Comparing faculty to national estimates describe how privileged the upbringings of faculty were relative to the general public, and comparisons to doctoral recipients describes how the socioeconomic roots of faculty differ from the broader set of their doctoral peers.

Data on the educational attainment of adults are drawn from the American Community Survey of the US Census, and information on family's educational attainment among Ph.D. recipients is drawn from the NSF SED. While historical data on undergraduate and graduate achievement is available from the NSF[70], it does not report demographics as percentages of the adult population or describe any earlier education levels as we require. Before 1993, the Census recorded educational attainment as years of schooling, and completion (or not) of high school or bachelor's degree[71]. More recent data recorded educational attainment as highest degree held among adults 25 years and older. In our study, we refer to both. Using the earlier Census data, we compare parents' college completion among faculty to college completion rates in the adult US population. When we require information on degree earned, we compare respondents with the Census dataset closest to the year faculty were born that records this information.

Our estimates of US adult Ph.D. completion (from 1993 at the earliest) are likely higher than the true Ph.D. completion rate in the year a respondent was born (which was 1967, on average), due to rising educational attainment in the United States. Thus, our comparisons of faculty parents with the general public may in fact underestimate academia's overrepresentation of highly educated families. Data on the highest degree attained by parents of Ph.D. recipients is available for select years from 1993 to 2018 from the NSF[69]. Details on how many respondents matched with a given year of data are provided in Supplementary Note B.

As childhood household income is also predictive of educational attainment, we construct proxy measures of childhood income levels and geographic diversity among current faculty. The ideal dataset would be individual-level household income and location during the childhoods of faculty. Because it is difficult for survey respondents' to estimate their parents' income retrospectively, we approximate these quantities using more easily recalled information. We link zip code responses to the closest publicly available IRS tax release (1998–2018) when faculty were children (adjusted to 2020 US $). Proxying individual SES based on aggregate zip code-level data is a widely used approach[72,73]. However, the geographic proxy may suffer from some mismeasurement because zip codes often span heterogeneous populations[74]. To measure whether faculty come from rural or urban areas, we use US Census data from 1990 to 2010, linked to faculty responses by their year of birth[75], which records how many people live in urban or rural areas in a given zip code. We labelled zip codes as either rural or urban based on whether the majority of the population in that zip code lived in a rural or urban area. As with our measure of income, this approach may suffer from mismeasurement due to the underlying heterogeneity of a zip code. These measures of childhood household income and geographic diversity, when taken together with self-reported information on parental education, speak to how faculty SES shapes academic careers.

Finally, to assess whether SES impacts job placement within academia, we compare the previously described measures with the institutional prestige of a respondent's current faculty appointment and Ph.D.-granting institution. For most of the disciplines considered here, we refer to 2017–2020 college rankings from USNWR, which are provided for most Carnegie-classified R1 and R2 doctoral universities. Because Anthropology departments are not ranked by USNWR, we use their R rank in the 2010 NRC Anthropology rankings. To account for differences in the number of departments across disciplines, we rescale each ranking to the unit interval. Where faculty were employed at institutions not ranked by USNWR or NRC (11.1% of responses), we coded their ranking as missing and excluded them from our analysis of institutional prestige.

**Reporting summary.** Further information on research design is available in the Nature Research Reporting Summary linked to this article.

## Data availability
Given the sensitive nature of the information provided by respondents to the survey, the underlying data cannot be de-identified in a way that would protect respondents' privacy while also preserving the data's utility for reanalysis or reuse. For these reasons, data can only be made available under an appropriate Institutional Review Board-approved data-sharing agreement.

## Code availability
All analyses done in Python. Code is available at https://github.com/allisonmorgan/faculty_ses.

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

## Acknowledgements

We thank E. Colunga, B. Donovan, B. Fosdick, T. Harris-Tryon, H. Lamothe, R. Manduca, J. Pyne, S. Spielman, K. Spoon, J. Van Cleve and S. Way for helpful comments, and all survey participants for providing their valuable opinions and time. This work was supported in part by the National Science Foundation (award No. SMA 1633791 to A.C.M., A.C., D.B.L. and M.G.), Air Force Office of Scientific Research (award No. FA9550-19-1-0329 to A.C., D.B.L., M.G. and N.L.), National Science Foundation Graduate Research Fellowship (award No. DGE 1650115 to A.C.M.) and California Center for Population Research at UCLA (CCPR), which receives core support from the Eunice Kennedy Shriver National Institute of Child Health and Human Development (NICHD) (grant No. P2C-HD041022 to J.E.B.). The funders had no role in study design, data collection and analysis, decision to publish or preparation of the manuscript.

Recent work in several fields of science has identified a bias in citation practices such that papers from women and other minority scholars are under-cited relative to the number of such papers in the field[76–80]. Here we sought to proactively consider choosing references that reflect the diversity of the field in thought, form of contribution, gender, race, ethnicity and other factors. First, we obtained the predicted gender of the first and last author of each reference by using databases that store the probability of a first name being carried by a woman[80,81]. By this measure (and excluding self-citations to the first and last authors of our current paper), our references contain 27.8% woman (first)/woman (last), 9.5% man/woman, 16.6% woman/man and 46.1% man/man. This method is limited in that (1) names, pronouns and social media profiles used to construct the databases may not, in every case, be indicative of gender identity, and (2) it cannot account for intersex, non-binary or transgender people. Second, we obtained predicted racial/ethnic category of the first and last author of each reference by databases that store the probability of a first and last name being carried by an author of colour[82,83]. By this measure (and excluding self-citations), our references contain 10.8% author of colour (first)/author of colour (last), 16.5% white author/author of colour, 14.3% author of colour/white author and 58.4% white author/white author. This method is limited in that (1) the names and Florida Voter Data used to make the predictions may not be indicative of racial/ethnic identity, and (2) it cannot account for Indigenous and mixed-race authors, or those who may face differential biases due to the ambiguous racialization or ethnicization of their names. We look forward to future work that could help us to better understand how to support equitable practices in science.

## Author contributions

A.C.M. and A.C. devised analysis, and A.C.M., A.C. and J.E.B. wrote the manuscript. M.G., A.C., D.B.L., A.C.M. and N.L. designed and conducted the survey. M.G., N.L. and A.C.M. collected data and performed data validation. A.C.M. and N.L. analysed data. All authors finalized the manuscript.

## Competing interests

The authors declare no competing interests.

## Additional information

**Correspondence and requests for materials** should be addressed to Allison C. Morgan or Aaron Clauset.

# Reporting Summary

## Statistics

For all statistical analyses, confirm that the following items are present in the figure legend, table legend, main text, or Methods section.

| n/a | Confirmed | |
|---|---|---|
| ☐ | ☒ | The exact sample size (*n*) for each experimental group/condition, given as a discrete number and unit of measurement |
| ☒ | ☐ | A statement on whether measurements were taken from distinct samples or whether the same sample was measured repeatedly |
| ☐ | ☒ | The statistical test(s) used AND whether they are one- or two-sided<br>*Only common tests should be described solely by name; describe more complex techniques in the Methods section.* |
| ☐ | ☒ | A description of all covariates tested |
| ☒ | ☐ | A description of any assumptions or corrections, such as tests of normality and adjustment for multiple comparisons |
| ☐ | ☒ | A full description of the statistical parameters including central tendency (e.g. means) or other basic estimates (e.g. regression coefficient) AND variation (e.g. standard deviation) or associated estimates of uncertainty (e.g. confidence intervals) |
| ☐ | ☒ | For null hypothesis testing, the test statistic (e.g. *F*, *t*, *r*) with confidence intervals, effect sizes, degrees of freedom and *P* value noted<br>*Give P values as exact values whenever suitable.* |
| ☒ | ☐ | For Bayesian analysis, information on the choice of priors and Markov chain Monte Carlo settings |
| ☒ | ☐ | For hierarchical and complex designs, identification of the appropriate level for tests and full reporting of outcomes |
| ☒ | ☐ | Estimates of effect sizes (e.g. Cohen's *d*, Pearson's *r*), indicating how they were calculated |

*Our web collection on statistics for biologists contains articles on many of the points above.*

## Software and code

Policy information about availability of computer code

| | |
|---|---|
| Data collection | No software was used for data collection. |
| Data analysis | Publicly available Python (3.9.1) libraries for statistics and visualization: matplotlib (3.3.3), numpy (1.19.5), pandas (1.2.0), scipy (1.6.0), seaborn (0.11.1), statsmodels (0.12.1), and geopandas (0.8.2). All analysis code available at: https://github.com/allisonmorgan/faculty_ses |

For manuscripts utilizing custom algorithms or software that are central to the research but not yet described in published literature, software must be made available to editors and reviewers. We strongly encourage code deposition in a community repository (e.g. GitHub). See the Nature Portfolio guidelines for submitting code & software for further information.

## Data

Policy information about availability of data

All manuscripts must include a data availability statement. This statement should provide the following information, where applicable:
- Accession codes, unique identifiers, or web links for publicly available datasets
- A description of any restrictions on data availability
- For clinical datasets or third party data, please ensure that the statement adheres to our policy

Given the sensitive nature of the information provided by respondents to the survey, the underlying data cannot be de-identified in a way that would protect respondents' privacy while also preserving the data's utility for reanalysis or reuse. For these reasons, data can only be made available under an appropriate IRB-approved data sharing agreement.

# Field-specific reporting

Please select the one below that is the best fit for your research. If you are not sure, read the appropriate sections before making your selection.

☐ Life sciences ☒ Behavioural & social sciences ☐ Ecological, evolutionary & environmental sciences

For a reference copy of the document with all sections, see nature.com/documents/nr-reporting-summary-flat.pdf

# Behavioural & social sciences study design

All studies must disclose on these points even when the disclosure is negative.

| | |
|---|---|
| Study description | We conducted a survey of professors in PhD-granting departments in the U.S. across STEM, social science, and the humanities, and provide a quantitative description. |
| Research sample | In total, 8,009 faculty responded to our survey (out of 46,692 surveyed; 17.2%). Of those, 7,204 faculty provided information on a parent's level of highest education (89.9% of respondents) and 4,807 provided the ZIP code in which they grew up (60.0%). Faculty that provided either parents' education or zip code for our analysis are generally representative with respect to their populations. Women responded at slightly higher rates than expected, which may imply a slight upward bias in our results in parents' education. In our sample, women are somewhat more likely to come from highly educated families. Respondents came from 2,494 unique ZIP codes across the U.S. Most ZIP codes were represented by only one respondent (76.9%). Questions about degree of parental support were asked towards the end of the survey (85.9% of respondents). Those respondents had similar demographics to those who provided parental education information, but skewed towards being from slightly less prestigious institutions in History, and slightly more in Sociology. More detail provided in Supplementary Information. |
| Sampling strategy | The sample frame was assembled by drawing a set of PhD-granting departments from the lists maintained by the U.S. News & World Report for history and business, and the Computing Research Association for computer science. The sample frames for Anthropology, Biology, History, Physics / Astronomy, Psychology, and Sociology are based on a comprehensive database of employment records for tenure-track faculty at U.S. PhD granting institutions that Academic Analytics–the academic consultancy that collected and maintains these faculty records provided for this research. Then, email addresses listed in online public directories were collected for faculty in our sample frame using both automated methods and Mechanical Turk. |
| Data collection | Relevant survey questions were as follows:<br><br>* ``In what year were you born?'' Drop down of years from 1916 to 1996.<br>* ``During the first 18 years of your life, did your family rent the home in which you lived, or did your family own it (even if supported by a mortgage)?'' Options were ``We rented a home during all or most of the first 18 years of my life,'' ``We rented and owned a home about equally often,'' ``We owned a home during all or most of the first 18 years of my life,'' or ``Don't know''<br>* ``Where did you live during the first 18 years of your life? If you lived in the U.S., please let us know in which ZIP code you live the longest.'' Open text box.<br><br>* ``Now please think of your parents or legal guardians during the first 18 years of your life and answer the following questions about them. If you grew up with just one parent or legal guardian, please select `Not applicable' for `Parent 2'.''<br>- ``What are their genders?'' Options were ``Male,'' ``Female,'' or ``Other identity.''<br>- ``What is their highest level of education?'' Options were ``Elementary: 0-4 years,'' ``Elementary: 5-8 years,'' ``High school: 1-3 years,'' ``High school: 4 years,'' ``College: 1-3 years,'' ``College: 4 or more years,'' ``Master's or professional degree,'' ``Doctoral degree,'' ``Don't know,'' or ``Not applicable / Rather not say.''<br>- ``What best describes their employment status during all or most of the first 18 years of your life?'' Options were ``Employed,'' ``Not employed: stay-at-home parent,'' ``Not employed: could not find job,'' ``Not employed: other reason (e.g. retired, illness, ...),'' ``Don't know or something else,'' or ``Not applicable / Rather not say.''<br><br>* ``What is your gender?'' Options were ``Male,'' ``Female,'' ``Other identity,'' or ``Prefer not to say.''<br>* ``What is your race or origin? Please select one or more responses.'' Options were ``White,'' ``Hispanic, Latino, or Spanish origin,'' ``Black or African American,'' ``Asian,'' ``American Indian or Alaska Native,'' ``Native Hawaiian or other Pacific Islander,'' ``Some other race or origin,'' or ``Prefer not to say.''<br>* ``Please rate support and encouragement you received for your academic career from people below, on a scale from 1 (none at all) to 5 (a lot). If some of those people were not present in your life, please choose Not applicable.'' Category analyzed was ``your parents'' with options 1--5 and ``Not Applicable.''<br><br>Survey was administered via emails, and recorded over the web. No blinding in response to experimental conditions were performed. |
| Timing | Our survey was conducted over three years, from Summer 2017 to Fall 2020. |
| Data exclusions | Any survey respondents who could not be identified from our frame were dropped. Otherwise no responses were excluded. |
| Non-participation | In total, 8,009 faculty responded to our survey (out of 46,692 surveyed; 17.2%). Of those, 7,204 faculty provided information on a parent's level of highest education (89.9% of respondents) and 4,807 provided the ZIP code in which they grew up (60.0%). |
| Randomization | All survey respondents received the same instruments. Our research questions did not require randomization. |

# Reporting for specific materials, systems and methods

We require information from authors about some types of materials, experimental systems and methods used in many studies. Here, indicate whether each material, system or method listed is relevant to your study. If you are not sure if a list item applies to your research, read the appropriate section before selecting a response.

## Materials & experimental systems

| n/a | Involved in the study |
|-----|----------------------|
| ☒ ☐ | Antibodies |
| ☒ ☐ | Eukaryotic cell lines |
| ☒ ☐ | Palaeontology and archaeology |
| ☒ ☐ | Animals and other organisms |
| ☐ ☒ | Human research participants |
| ☒ ☐ | Clinical data |
| ☒ ☐ | Dual use research of concern |

## Methods

| n/a | Involved in the study |
|-----|----------------------|
| ☒ ☐ | ChIP-seq |
| ☒ ☐ | Flow cytometry |
| ☒ ☐ | MRI-based neuroimaging |

## Human research participants

Policy information about studies involving human research participants

| | |
|---|---|
| Population characteristics | In total, 7,204 faculty provided information on a parent's level of highest education (15.4% of survey frame) and 4,807 provided the U.S. ZIP code in which they grew up (10.3%). Across the eight disciplines surveyed, women represented between 19.6% (Physics / Astronomy) and 53.7% (Sociology) of responses. Average year of birth of respondents was 1967. |
| Recruitment | Each individual received one email reminder. All participants were included in a drawing for a cash lottery. Amounts varied by survey wave: one $1,000 payment for Computer Science, $500 for Business and History faculty, $500 for Anthropology, Physics / Astronomy, Psychology, and Sociology faculty, and $250 for Biology faculty. To reduce the burden on participants, our survey was divided into two parts. The first part took 1-2 minutes to complete and asked the most important questions of interest for our study, including participants' year of birth, childhood ZIP code, and information on their parents' education and employment. After completing the first part, participants were told that they can continue to the second part, which asked about parental support for their careers. |
| Ethics oversight | University of Colorado Boulder IRB (Protocol #16-0441) |

Note that full information on the approval of the study protocol must also be provided in the manuscript.

