## [Peer Review File. · Nature Human Behaviour]

Peer Review Information

Journal: Nature Human Behaviour

Manuscript Title: Socioeconomic Roots of Academic Faculty

Corresponding author name(s): Allison C. Morgan and Aaron Clauset

Reviewer Comments & Decisions:

Decision Letter, initial version:

23rd September 2021

Dear Dr Morgan,

Thank you once again for your manuscript, entitled "Socioeconomic Roots of Academic Faculty", and for your patience during the unusually lengthy peer review process.

Your Article has now been evaluated by 3 referees. You will see from their comments copied below that, although they find your work of considerable potential interest, they have raised quite substantial concerns. In light of these comments, we cannot accept the manuscript for publication, but would be interested in considering a revised version if you are willing and able to fully address reviewer and editorial concerns.

We hope you will find the referees' comments useful as you decide how to proceed. If you wish to submit a substantially revised manuscript, please bear in mind that we will be reluctant to approach the referees again in the absence of major revisions. We are committed to providing a fair and constructive peer-review process. Do not hesitate to contact us if there are specific requests from the reviewers that you believe are technically impossible or unlikely to yield a meaningful outcome.

Reviewer #2 raises several methodological issues, including the representativeness of the respondent sample given low response rate, and ZIP-code level income data, and we ask that you address them in your revision, also acknowledging the limitations of your data where addressing these concerns fully would not be possible. Reviewers #1 and #3 point to several issues with the framing of your article, as well as with argumentation and discussion of your findings in the context of the existing sociological literature. Given the nature of your research question, we believe that these issues are important and should be addressed via a collaboration with a sociologist specializing in stratification and social mobility.

If you wish to submit a suitably revised manuscript we would hope to receive it within 6 months. We

understand that the COVID-19 pandemic is causing significant disruptions which may prevent you from carrying out the additional work required for resubmission of your manuscript within this timeframe. If you are unable to submit your revised manuscript within 6 months, please let us know. We will be happy to extend the submission date to enable you to complete your work on the revision.

- Include a "Response to the editors and reviewers" document detailing, point-by-point, how you addressed each editor and referee comment. If no action was taken to address a point, you must provide a compelling argument. This response will be used by the editors to evaluate your revision and sent back to the reviewers along with the revised manuscript.
- Highlight all changes made to your manuscript or provide us with a version that tracks changes.

[REDACTED]

Thank you for the opportunity to review your work. Please do not hesitate to contact me if you have any questions or would like to discuss the required revisions further.

Sincerely,

Arunas Radzvilavicius, PhD
Editor
Nature Human Behaviour

Reviewer expertise:

Reviewer #1: sociology, organizational studies, inequalities

Reviewer #2: life course sociology, educational mobility

Reviewer #3: sociology of education, inequality in higher education

REVIEWER COMMENTS:

Reviewer #1:
Remarks to the Author:

Review of Socioeconomic Roots of Academic Faculty

In a pre-pandemic job market dinner with a potential candidate, the people at the table realized at some point in the conversation that all four of us at the table were children of academics. My father was a professor. In college I lived in a group house composed of almost all faculty kids. We are now mostly all faculty. The cohort ahead of me in graduate school included someone whose father was president of the American Sociological Association. One of my colleagues has two parents who are professors. His father works at our institution, and his sibling is also an academic. This paper makes visible at the systemic level what many of those who live in this world experience in our daily lives.

Overall, the findings of the paper are importance, but the prose and the framing need a bit more work.

Some of the findings that I found interesting (if sadly utterly unsurprising):

1. The general finding that nearly a quarter of faculty report PhD parents
2. That children of PhDs are over-represented at elite institutions
3. The implications these patterns have for the racial diversification of the faculty, as few Black and Hispanic are PhD holders, thus meaning their children are less likely to benefit from this advantage

4. The fact that

“the importance of having PhD parents appears so great that the rate of having them nearly doubles across the transition from completing a PhD to obtaining a faculty job (11.8% versus 22.2%; Table I).” (The implications of this finding could be highlighted more, as it seems there is a loss of talent from the pipeline in the transition to the first job. I see this with my own graduate students, as those with PhD parents are better situated psychologically, financially, and logistically to ride out the difficulties of the academic job market. The economic and professional precarity introduced by the pandemic is likely to increase the hemorrhage of those without highly educated parents out of the academia).

The paper unfortunately reads a bit as written by a group of computer scientists who have taken a crash course in sociology. It could be improved by perhaps adding a sociological stratification scholar to the team to revise the framing of the paper, or through getting some more feedback from sociologists. Some places where the framing/prose/argumentation are problematic:

- The second paragraph of the introduction is problematic. It reads as overly individualistic in terms of attributing the lower educational achievements of poor parents to their failures as parents, neglecting a vast sociological literature on the way poor kids are assigned to worse schools, and the inequalities in the treatment of children by class and race within schools. See, for example, Lewis and Diamond, *Despite the Best Intentions*. Lareau and Calarco would almost certainly see their work as misinterpreted by the second sentence of this paragraph.
- I’m not sure whether Anthony Jack’s book actually documents that lower-income students are more likely to come from privileged high school prep programs. Verify.
- The 3rd paragraph needs to be rewritten by a sociologist who specializes in this area of research. It is evident in the clumsiness here of dealing with this literature that no one with expertise in these research

traditions is on the research team.

- The paper moves inconsistently between a structural/constructionist/sociology of science framework and a biological/individualist framing. The paper has 5 authors and feels like it, in that there isn't a consistent argumentative voice guiding the framing and interpretation. The first perspective is the stronger thread. As the contributions of this paper are empirical, not theoretical, it doesn't have to be theoretically original, but a smooth authorial voice situated in a consistent way within the literature would be good.

There are several places where the prose is confusing. The paper could do with careful cleanup, with attention to explaining the findings as clearly as possible.

Abstract

It is clear how the socioeconomic roots of academic factor impact the race of the faculty, and, by the end of the article, the geographical implications are clear. The gender implications are less clear. I stumbled on this claim.

Also, it took me a couple of readings to decode the last line of the abstract due to complex sentence construction.

Include the years that the study examines (2017 to 2020) in the abstract and very early in the paper. It is buried at present.

Data and Methods

WHEN the data were collected should be front and center in this section as should the response rate.

I found the discussion on page on "obtaining precise bounds" to be confusing.

Results

In the first paragraph of results, there is a fair amount of work to move from what is presented in Table 1 and the percentages presented in the text. Maybe possible to restructure the table to show both the disaggregated data that is there, and how it aggregates up to what is discussed in the text?

Maybe add subheadings to results, depending on the formatting requirements of the journal?

I'm really interested in what is going on with women having more educated mothers than men. Make sure to flag this for more research.

I was confused on page 5 in the discussion in the first paragraph:

Faculty who reported at least one of their parents holds a PhD traveled about the same distance from home (873 versus 835 miles; Kolmogorov-Smirnov, $D = 0.0$, $N = 7209$, $p > 0.01$), but are slightly younger at the start of the tenure-track job than those faculty whose parents do not hold a PhD (33.1 versus 33.6 years old; $t = -3.0$, $N = 6951$, $p < 0.01$).

The discussion of race on page 5, second column, is very important, but the discussion is currently unclear.

Gender comes up again on page 5, in the finding that 25.5% of White women have a PhD parent versus 14.6% of Black women (Table S3). And then again on page 6, with the finding that Black women are less likely to have PhD parents than Black men. The authors do flag these as interesting findings for more investigation, but I think the need more some intersectional research here can be flagged even more boldly.

On page 6 in the paragraph on PhD parents and academic hierarchy, there is a suggestion at the end of the paragraph that “academic hiring tends to devalue faculty of lower socioeconomic outstanding.” I think that is true, but my guess is that this probably operates as much or more through status of PhD program than direct discrimination. From my observations, I think there are a lot of mechanisms at play:

1. Children of PhDs are dissuaded by parents from going to graduate school if they cannot get into elite programs with full funding (e.g., if they cannot get into programs that generally track to elite jobs).
2. Children of PhDs are often deliberately sent to SLACs that have a history of producing PhDs (that track into elite jobs) and are often steered away from schools that pride themselves on producing those successful in business and other professional fields. My PhD friends tend to send their kids to Oberlin, Reed, Wesleyan, University of Chicago, and similar schools, for example.
3. Parents of PhDs are more likely to be willing and able to provide substantial financial and other investments when their offspring are in graduate school – particularly in the year or years it takes to secure an academic job at the end of graduate school, enabling them to turn down less good jobs, delay going on the market until the c.v. is competitive, or survive for a year or so without employment until a job is secured. (e.g., my parents paid for my son’s childcare, my credit card bills and holiday travel, and my father helped me produce the figures for my first book). These kinds of highly consequential investments go well beyond generic support and encouragement. I do not think these investments are unusual. I mentor undergraduates with PhD parents and it is clear that their parents advise them on research design and providing deeply substantive input on their projects.
4. Children of PhDs are familiar with the sorts of people who will be their dissertation advisors, colleagues, on their tenure committees. The deep understanding that they have about how to please authorities with certain tastes and values is, I think, an impossible to quantify advantage.
5. PhD parents are situated to channel their kids toward academic from the selection of AP courses in high school, through college choice, college major, research experiences, graduate school selection, strategies to navigate graduate school, key resources and forms of capital to survive and thrive graduate school, strategies to navigate the job market, etc.

These mechanisms are, of course, vastly too fine-grained for this paper. I guess this is all by way of saying that more research on the specific parenting practices and mechanisms that produce these outcomes is suggested by this research.

The first two paragraphs on the top of page 7 are confusing.

Conclusion

This point is very important but could be stated more clearly. The wording is convoluted:

"In the context of broader racial inequality in wealth and educational attainment with the U.S., academia's overrepresentation of inherited advantages, i.e., the importance of parental characteristics for a professor's current employment and placement, represents a fundamental limit to its racial diversity (Fig. 6)."

The long footnote on page 8 genetic variation is unnecessary, as the paper is not about explaining the patterns, but about documenting them. Even including genetic variation in the list of factors in the mix on the bottom of page/top of page 8 just inserts a bomb into the middle of the paper that can lead it be conveyed by the media as, "scholars argue that PhD parents produce faculty kids because they have better genes!" This paper just doesn't need to engage in this debate.

The point that the socioeconomic background may influence the type of scholarship that faculty pursue is important. Note that there is extensive research in sociology in the areas of race, gender, and sociology of science that identity and structural location influences knowledge production. Some of this work could be cited at the end of the first paragraph of the second column on the top of page 8.

Details

Abstract: I think a comma after "For example" in the 4th line would be helpful

Introduction:

First paragraph: Cut "comparatively" from first paragraph. Rephrase last sentence

First page, second column: subject/verb agreement – processes ARE

Reviewer #2:

Remarks to the Author:

Using survey data obtained from 7204 professors in eight fields including STEM, social science, and the humanities, linked (via ZIP codes) to aggregate information from national public datasets, this article examines the socioeconomic background, race, and gender composition of faculty in the U.S. Examining faculty exclusivity is a worthwhile endeavor. However, several issues detract from the value and contribution of this research.

A central motivation for the study, stated in the abstract and again on p. 2, appears to be that a "lack of socioeconomic diversity is likely to deeply shape the type of scholarship and scholars that faculty produce and train." However, only two citations are given in support of this claim, a study of "combating anti-blackness in the ai community" and a book that does not appear to be focused on faculty, but instead more generally on "the power of diversity" in "groups, firms, schools, and societies." Moreover, the conclusion (p. 8) admits the lack of empirical evidence regarding the effect of background on scholarship: "...it remains unknown how socioeconomic background influences the type of scholarship

that faculty pursue.” The explosion of research on race, racism, social justice, gender discrimination, and inequality, etc., across disciplines, would appear to belie the implication (not stated overtly) that the high socioeconomic background of faculty would lead to conservative scholarship, diverting attention from such topics.

Almost 1 of 5 references date from the 1990’s or earlier. Surely more recent work documents intergenerational transmission and the difficulties of those from working class backgrounds in higher educational and professional contexts. The authors may find *Paying for the Party* (Hamilton and Armstrong 2013) instructive.

Not surprisingly, given the well-established linkages between family background and educational attainment, and the near-universal requirement that faculty in Ph.D.-granting departments have Ph.D.’s, U.S. faculty are found to be many times more likely than the general population to have parents who are Ph.D. recipients. It is also to be expected that faculty with Ph.D. parents would receive more support from their parents for their academic careers, given parents’ greater familiarity with academia. These findings are not news. The findings do indicate a clear disadvantage for racial/ethnic minorities who are less likely to have parents with doctorates (Table S3).

The comparison in Table 1, showing that 11.8 percent of Ph.D. recipients have parents with Ph.D.’s, compared to 22% of those with a faculty job purportedly illustrates the great advantage of having a Ph.D. level parent. However, many of those who did not enter academia (for example, in highly technical STEM fields) after receiving a Ph.D., may have been aiming for higher paying non-academic employment in industry, think tanks, etc., and not even seeking jobs in higher education.

It is noted (p. 6) that “nearly a third of faculty at top ranked universities across all eight fields report that one of their parents holds a PhD” (29.5%). Looked at another way, might one argue that the majority of even the most elite faculty (~60%) are upwardly mobile, more highly educated than their parents and without the socialization and other advantages that a parent with a Ph.D. would confer?

With respect to the methods, the 17% response rate is cause for concern (even though this may be typical of online surveys). It is stated (p. 2) that “..respondents are generally representative of their fields by institutional prestige and faculty rank,” and Table S1 shows few differences. But given the very low response rate, more attention should be given to this issue. The extent to which those who chose to respond to the survey were different in terms of socioeconomic origins from other faculty who did not respond is not discussed.

Given considerable evidence that college degrees are more important in the labor force than years of college attendance (see work by James Rosenbaum), it is unclear why the survey measure of parental education distinguished less than four years and four years of college education. Instead, it would be more illuminating to inquire about college degree receipt (BA or BS) and some college without degree. Also, the measure of race departs from the current standard. Since persons of Hispanic ethnicity can be any race, the US Census and most surveys now ask first, Are you Hispanic or Latino (yes-no) followed by What is your race (White, Black or African American, etc.).

It is unclear why the coefficients for parents’ highest degree in Model III, Table II, are all negative. Shouldn’t those for college, masters and Ph.D. lead to higher institutional prestige than some college?

The authors are aware that the aggregate income data is problematic, given the heterogeneity of ZIP codes. However, it is to be expected, given the education of parents, that parental incomes of faculty members would be greater than the general population.

In sum, in view of these problems, I do not believe that the findings presented here represent a significant enough advance in knowledge to be of interest to scholars in my field or in other academic field

Reviewer #3:

Remarks to the Author:

Thanks for the opportunity to read this interesting and valuable analysis. Overall this paper makes important contribution to broader examinations of inequality in academia by focusing on faculty members' socioeconomic and intersectional backgrounds, which have been comparatively neglected in previous research. Overall I have few suggestions or critiques to make, which I hope will be helpful to the authors as they revise.

Framing

The authors note in the abstract that "Tenure-track faculty play a special role in society..." However, it's not only TT faculty who make these contributions, but also lecturers, contingents/adjuncts, etc. TT faculty have economic and employment privileges and these jobs are much harder to get. I think this distinction is important to clarify in the framing. I agree with the focus on TT, but think there's more to be parsed out here.

Similarly, the article focuses only on those who have obtained TT jobs, leaving aside adjuncts or other contingent faculty and those who left academia, which should be acknowledged. Again, I don't see this sampling choice as unreasonable (especially given the comparisons drawn among PhD holders) but think the authors are missing some important contextual dynamics or forms of inequality that shape these outcomes.

Finally, it seems like there should be some discussion or at least note of racism, classism, and gender (and the intersections thereof) facing graduate students and faculty members that may drive down both hiring rates and intention to persist, as well as placements and tenure among other achievements in ways that, as the authors note, are also related to SES.

Clarifying wording: The authors' wording is unclear in some places.

P1: "Working class or poor parents are less likely to enroll their children in organized activities or allow their children to question establishments..." – This is true, but it should be clear that this is at least in part structural—less access, fewer resources.

P1: This sentence is a little confusing—"The association between the economic circumstances of parents and their children is often less strong (upward social mobility) for children who complete college than for

those who do not [17, 18].” – does this include continuing generation children?

P1: The authors write that, “Occupations specifically play an important role in the intergenerational transmission of class [25], and an association between children and their parents’ occupations (sometimes called microclass reproduction or inequality) has persisted for decades [26]. This pattern is not itself problematic, but can be symptomatic of social constraints on personal choice that systematically limit social mobility [27]” – Given the discussion so far, I’m not sure why is it not problematic.

P2: “Nevertheless, it is known that faculty are increasingly representative of highly educated families [28]...” – increasingly over time? That would surprise me. This citation is also a bit dated. I don’t think this particular point is needed, given that this is a big part of what the article seeks to establish in the authors’ own findings.

Other minor issues

Is it the case that children of lower-income families are less likely to complete elementary school in the US (as the authors claim on P1)? The citation given is cross-national, which seems either not relevant or unclear—the article appears to be framed around US SES dynamics, not international. If that’s not the case, the authors should indicate that they include faculty respondents from cross-national backgrounds. 1999 also seems like a somewhat distant point in the past for this kind of citation.

In the discussion of Fig. 6, what accounts for the much higher and much more volatile rates of PhD attainment among Asian adults?

The authors make strong points about how race/ethnicity and SES, in particular, are related in these patterns. This seems worth reiterating at the start of the conclusion where the main findings are recapped.

Author Rebuttal to Initial comments

Response to Review 1

Reviewer #1: sociology, organizational studies, inequalities

In a pre-pandemic job market dinner with a potential candidate, the people at the table realized at some point in the conversation that all four of us at the table were children of academics. My father was a professor. In college I lived in a group house composed of almost all faculty kids. We are now mostly all faculty. The cohort ahead of me in graduate school included someone whose father was president of the American Sociological Association. One of my colleagues has two parents who are professors. His father works at our institution, and his sibling is also an academic. This paper makes visible at the systemic level what many of those who live in this world experience in our daily lives.

Overall, the findings of the paper are important, but the prose and the framing need a bit more work.

We thank the reviewer for their positive assessment of our study, and for highlighting the visibility that our analysis brings to the systemic characteristics of academia. And, we appreciate the thoughtful suggestions on improving the manuscript. We believe addressing them has substantially improved the overall framing, discussion, and relationship with past relevant work in this area.

Some of the findings that I found interesting (if sadly utterly unsurprising):

- 1. The general finding that nearly a quarter of faculty report PhD parents**
- 2. That children of PhDs are over-represented at elite institutions**
- 3. The implications these patterns have for the racial diversification of the faculty, as few Black and Hispanic are PhD holders, thus meaning their children are less likely to benefit from this advantage**
- 4. The fact that “the importance of having PhD parents appears so great that the rate of having them nearly doubles across the transition from completing a PhD to obtaining a faculty job (11.8% versus 22.2%; Table I).”**

(The implications of this finding could be highlighted more, as it seems there is a loss of talent from the pipeline in the transition to the first job. I see this with my own graduate students, as those with PhD parents are better situated psychologically, financially, and logistically to ride out the difficulties of the academic job market. The economic and professional precarity introduced by the pandemic is likely to increase the hemorrhage of those without highly educated parents out of the academia).

We agree that the implications of this finding could be highlighted further. In the revised manuscript, we have drawn out and more clearly highlighted these implications in the conclusion.

The paper unfortunately reads a bit as written by a group of computer scientists who have taken a crash course in sociology. It could be improved by perhaps adding a sociological stratification scholar to the team to revise the framing of the paper, or through getting some more feedback from sociologists.

We fully agree that the framing should be improved, especially from the perspective of better integrating the ideas and results into the social inequality and stratification literature. In this

revised version, Prof. Jennie Brand (Sociology at UCLA) has joined the paper as a coauthor, bringing her substantial expertise in social stratification, social inequality, education, social demography, and quantitative methods. Brand is a well-known scholar of social stratification and inequality. She is currently Chair of the Inequality, Poverty, and Mobility Section of the American Sociological Association and a Member of the Board of Directors of the International Sociological Association Research Committee of Social Stratification and Mobility.

Dr. Brand's contributions have helped substantially reshape the narrative and interpretation of the results, and she has helped craft our responses to the reviews. We thank the reviewer for this suggestion.

Some places where the framing/prose/argumentation are problematic:

- **The second paragraph of the introduction is problematic. It reads as overly individualistic in terms of attributing the lower educational achievements of poor parents to their failures as parents, neglecting a vast sociological literature on the way poor kids are assigned to worse schools, and the inequalities in the treatment of children by class and race within schools. See, for example, Lewis and Diamond, *Despite the Best Intentions*. Lareau and Calarco would almost certainly see their work as misinterpreted by the second sentence of this paragraph.**

In the revised manuscript, we have substantially revised the first paragraph and fully rewrote the second paragraph. We eliminated the overly individualistic text and incorporated the sociological literature on social class and race and educational attainment. We incorporated Lewis and Diamond, as well as other research in the sociology of education. And, we revised the text on the contributions of Lareau and Calarco.

- **I'm not sure whether Anthony Jack's book actually documents that lower-income students are more likely to come from privileged high school prep programs. Verify.**

In the revised manuscript, we have removed this citation to Jack.

- **The 3rd paragraph needs to be rewritten by a sociologist who specializes in this area of research. It is evident in the clumsiness here of dealing with this literature that no one with expertise in these research traditions is on the research team.**

We have rewritten the third paragraph to focus on the stratification of graduate level education.

- **The paper moves inconsistently between a structural/constructionist/sociology of science framework and a biological/individualist framing. The paper has 5 authors and feels like it, in that there isn't a consistent argumentative voice guiding the framing and interpretation. The first perspective is the stronger thread. As the contributions of this paper are empirical, not theoretical, it doesn't have to be theoretically original, but a smooth authorial voice situated in a consistent way within the literature would be good.**

We appreciate the feedback that the authorial voice in the original manuscript was not as consistent as we'd intended. In the revised version, we have endeavored to resolve this issue throughout. In particular, the revised version moves away from a biological/individualistic framing, and we believe the revised framing now better articulates the scientific findings and their sociological context.

There are several places where the prose is confusing. The paper could do with careful cleanup, with attention to explaining the findings as clearly as possible.

We appreciate the encouragement to write more clearly, and the specific suggestions of places that need attention. We have made a variety of changes throughout the manuscript to improve the exposition around these points, and we provide some additional detail below.

Abstract

It is clear how the socioeconomic roots of academic factor impact the race of the faculty, and, by the end of the article, the geographical implications are clear. The gender implications are less clear. I stumbled on this claim.

We agree that the gender results are less significant than the results on race, and have removed them from the abstract in order to focus the reader's attention on the more interesting findings of the study. In the article itself, we have also revised the exposition around the gender results in order to make them more clear.

Also, it took me a couple of readings to decode the last line of the abstract due to complex sentence construction.

We agree that the original wording of this sentence in the abstract was too complex. In the revised version, we have rewritten this sentence to be more clear.

Include the years that the study examines (2017 to 2020) in the abstract and very early in the paper. It is buried at present.

Data and Methods

WHEN the data were collected should be front and center in this section as should the response rate.

We agree that this information is important to the interpretation of our work, and we apologize for not making this more clear in the initial draft. In the revised version, we have added the years of the survey to the abstract of the paper and the introduction, and we have added the response rate information to the introduction.

I found the discussion on page on “obtaining precise bounds” to be confusing.

We agree that this point is subtle, and that our initial wording was not as clear as intended. The problem arises from the way the two different sources of data were collected: the U.S. population data on PhD attainment gives the number of PhDs per capita, which is an individual-level measure, while respondents to our survey indicated whether at least one of their parents holds a PhD, which is a non-individual-level measure. As a result, the two rates are not precisely comparable because respondents will typically have two parents, one or both of whom could have a PhD. Furthermore, the probability that both parents hold a PhD is not the product of the individual probabilities because of homophily. The estimation of upper and lower bounds is methodologically necessary to reconcile these differences; they account for the two extremes of all pairs of parents holding PhDs and only ever one parent holding a PhD.

The technical details of trying to produce an accurate estimation of the true rate are not appropriate for the main text, but we do believe this point is still important enough to be discussed explicitly. In the revision, we have moved most of this discussion to the supplement, and expanded it in order to more clearly explain to the reader the form of the two sets of data

and what kinds of comparisons they allow.

Results

In the first paragraph of results, there is a fair amount of work to move from what is presented in Table 1 and the percentages presented in the text. Maybe possible to restructure the table to show both the disaggregated data that is there, and how it aggregates up to what is discussed in the text?

We have reworded this paragraph to more closely tie to the results in Table I.

Maybe add subheadings to results, depending on the formatting requirements of the journal?

We agree that subheadings would aid in the clarity of the manuscript. In the revised version, we have added several to the Results section.

I'm really interested in what is going on with women having more educated mothers than men. Make sure to flag this for more research.

We thank the reviewer for highlighting this finding, and agree that this result needs more attention. In the revised version, we revisit this empirical finding in the Discussion where we highlight its importance for future research.

I was confused on page 5 in the discussion in the first paragraph:

Faculty who reported at least one of their parents holds a PhD traveled about the same distance from home (873 versus 835 miles; Kolmogorov-Smirnov, $D = 0.0$, $N = 7209$, $p > 0.01$), but are slightly younger at the start of the tenure-track job than those faculty whose parents do not hold a PhD (33.1 versus 33.6 years old; $t = -3.0$, $N = 6951$, $p < 0.01$).

We agree that this sentence, as originally worded, was confusing. In the revised manuscript, we have split this paragraph and added further elaboration to each point, in order to provide more context for each specific finding.

In particular, we moved the result that faculty grew up in relatively urban settings closer to estimated income and home-ownership results. Because faculty from higher SES backgrounds may be more comfortable moving farther from home, or may navigate the transition into a tenure-track job more quickly, we have moved the results about geographic mobility and age into the section about modeling faculty placement.

The discussion of race on page 5, second column, is very important, but the discussion is currently unclear.

We apologize for the lack of clarity here in the original submission. Our estimate for the probability of becoming a faculty member, given that one's parents have a PhD, does not take into account race or ethnicity.

Given the dramatically lower rates of PhD attainment among Black and Hispanic U.S. adults ($P(\text{PhD parent})$), as well as the percentage of Black or Hispanic faculty ($P(\text{faculty})$), and, within

our survey, the differences in the rates of PhD parents among Black and Hispanic faculty ($P(\text{PhD parent} \mid \text{faculty})$), could lead us to believe the not race-conditioned estimate in the paper will be an overestimate for Black and Hispanic adults.

In order to estimate this quantity precisely though, we would need additional data that is not currently available in our study. For example, the Bureau of Labor Statistics does not currently produce an estimate of employment rates by sector and race ($P(\text{faculty} \mid \text{race})$). We have adjusted the text in this section to clarify this calculation, and to note more explicitly the calculation's utility in understanding how conditioning on race would likely change the outcome.

Gender comes up again on page 5, in the finding that 25.5% of White women have a PhD parent versus 14.6% of Black women (Table S3). And then again on page 6, with the finding that Black women are less likely to have PhD parents than Black men. The authors do flag these as interesting findings for more investigation, but I think the need more some intersectional research here can be flagged even more boldly.

We agree that these are very interesting findings and have added an explicit call out to this direction of research – how race, gender and social upbringing intersect to shape the careers of tenure-track faculty – to the end of the section “Relationships with Racial Diversity.”

We have also added a citation to McDaniel et al's “The Black Gender Gap in Educational Attainment: Historical Trends and Racial Comparisons” (2011) which looks at the gender gap in educational attainment (women outpacing men) among both Black and White Americans. We cannot say from our study why Black women faculty were less likely to come from families with PhD parents than Black men, but agree that it is an important direction of more research.

On page 6 in the paragraph on PhD parents and academic hierarchy, there is a suggestion at the end of the paragraph that “academic hiring tends to devalue faculty of lower socioeconomic outstanding.” I think that is true, but my guess is that this probably operates as much or more through status of PhD program than direct discrimination.

From my observations, I think there are a lot of mechanisms at play:

- 1. Children of PhDs are dissuaded by parents from going to graduate school if they cannot get into elite programs with full funding (e.g., if they cannot get into programs that generally track to elite jobs).**
- 2. Children of PhDs are often deliberately sent to SLACs that have a history of producing PhDs (that track into elite jobs) and are often steered away from schools that pride**

themselves on producing those successful in business and other professional fields. My PhD friends tend to send their kids to Oberlin, Reed, Wesleyan, University of Chicago, and similar schools, for example.

3. Parents of PhDs are more likely to be willing and able to provide substantial financial and other investments when their offspring are in graduate school – particularly in the year or years it takes to secure an academic job at the end of graduate school, enabling them to turn down less good jobs, delay going on the market until the c.v. is competitive, or survive for a year or so without employment until a job is secured. (e.g., my parents paid for my son’s childcare, my credit card bills and holiday travel, and my father helped me produce the figures for my first book). These kinds of highly consequential investments go well beyond generic support and encouragement. I do not think these investments are unusual. I mentor undergraduates with PhD parents and it is clear that their parents advise them on research design and providing deeply substantive input on their projects.
4. Children of PhDs are familiar with the sorts of people who will be their dissertation advisors, colleagues, on their tenure committees. The deep understanding that they have about how to please authorities with certain tastes and values is, I think, an impossible to quantify advantage.
5. PhD parents are situated to channel their kids toward academic from the selection of AP courses in high school, through college choice, college major, research experiences, graduate school selection, strategies to navigate graduate school, key resources and forms of capital to survive and thrive graduate school, strategies to navigate the job market, etc.

These mechanisms are, of course, vastly too fine-grained for this paper. I guess this is all by way of saying that more research on the specific parenting practices and mechanisms that produce these outcomes is suggested by this research.

We fully agree that there are many possible mechanisms that may be or are likely to be important in driving the large-scale empirical patterns we find in our study, and we also agree that a thorough articulation and exploration of these mechanisms is beyond the scope of this paper. We very much appreciate the reviewer’s constructive comments in this direction, which highlight the complexity and difficulty of getting to causal explanations of the general patterns we observe.

At the same time, we believe the overall framing and interpretation would be enhanced by some engagement with these ideas, now included in our revised Discussion. We now write: “PhD parents are uniquely positioned to offer guidance and support in children’s post-secondary studies, through the graduate application process, negotiating admission offers, progress in

graduate programs, and job market success. Deep understanding of these processes facilitates micro-class reproduction.”

The first two paragraphs on the top of page 7 are confusing.

We agree that these paragraphs benefited from greater clarity. We have reorganized and revised them in order to better communicate our findings about lower-SES faculty at elite institutions, and the association of higher-SES faculty with elite pedigree has been moved to the Discussion section, as evidence of a possible mechanism by which SES shapes academic placement.

Conclusion

This point is very important but could be stated more clearly. The wording is convoluted: “In the context of broader racial inequality in wealth and educational attainment with the U.S., academia’s overrepresentation of inherited advantages, i.e., the importance of parental characteristics for a professor’s current employment and placement, represents a fundamental limit to its racial diversity (Fig. 6).”

We thank the reviewer for the suggestion, and have reworded this sentence for clarity as follows: “... given broader racial inequality in educational attainment, academia’s overrepresentation of inherited advantages represents a fundamental constraint to increasing its racial diversity (Fig. 6).”

The long footnote on page 8 genetic variation is unnecessary, as the paper is not about explaining the patterns, but about documenting them. Even including genetic variation in the list of factors in the mix on the bottom of page/top of page 8 just inserts a bomb into the middle of the paper that can lead it be conveyed by the media as, “scholars argue that PhD parents produce faculty kids because they have better genes!” This paper just doesn’t need to engage in this debate.

We fully agree that the discussion of connections with behavioral genetics is likely to be unhelpfully distracting to some readers, and that it detracts from the more substantive contributions of the paper. We appreciate the reviewer pushing us in this direction. In the revised version, we have simply removed it and the associated references.

The point that the socioeconomic background may influence the type of scholarship that faculty pursue is important. Note that there is extensive research in sociology in the areas

of race, gender, and sociology of science that identity and structural location influences knowledge production. Some of this work could be cited at the end of the first paragraph of the second column on the top of page 8.

We thank both this reviewer and another reviewer for raising this concern.

To better explain these ideas to the reader, in the introduction of the manuscript, we have added references and discussion of three recent studies which quantitatively examine how researcher identity shapes research topics [Kozlowski et al. “Intersectional inequalities in science” *PNAS* (2021); Hofstra et al. “The Diversity-Innovation Paradox in Science” *PNAS* (2020); Hoppe et al. “Topic choice contributes to the lower rate of NIH awards to African-American/black scientists” *Science Advances* (2019)]. In our discussion, we draw upon these studies to point out some of the research topics that they find are often pursued by minoritized researchers (e.g., public health and gender discrimination).

We also draw a relationship to Bourdieu's “The specificity of the scientific field and the social conditions of the progress of reason” *Social Science Information* (1975) which explains how different scientific strategies (such as research topic choices) are related to social capital (referring to the differing resources of lower SES or POC faculty) and the investment / disinvestment of that capital.

We were attempting to highlight that despite understanding that social capital relates to research choices, it is difficult to identify quantitatively the set of unstudied research topics or to model how changes in representation might change the kind of discoveries or contributions that are made. These are hard but important counterfactual questions, and at this point, we as a field do not fully understand what we (as a community) are missing out on, nor has the field developed reliable ways to investigate these questions. By providing clear and quantitative assessments of current diversity, and its connection to socioeconomic background, our study represents a critical step toward achieving that larger goal.

We fully agree that these are important points to convey clearly to the reader. In the revised version, we have endeavored to better highlight that this is what we meant by “unknown.” We believe that addressing these points has greatly improved our paper's framing.

Details

Abstract: I think a comma after “For example” in the 4th line would be helpful

We agree and have added a comma to this sentence.

Introduction:

First paragraph: Cut “comparatively” from first paragraph. Rephrase last sentence.

We have removed the word comparatively, and have rewritten the last sentence of this paragraph as follows: “Analyses of the socioeconomic origins of faculty help us understand the social reproduction of the highest levels of academic attainment and scientific influence.”

First page, second column: subject/verb agreement – processes ARE

We thank the reviewer for flagging this. We have rewritten this whole paragraph.

Response to Review 2

Reviewer #2: life course sociology, educational mobility

Using survey data obtained from 7204 professors in eight fields including STEM, social science, and the humanities, linked (vis ZIP codes) to aggregate information from national public datasets, this article examines the socioeconomic background, race, and gender composition of faculty in the U.S. Examining faculty exclusivity is a worthwhile endeavor.

We thank the reviewer for recognizing the scientific value of our study.

However, several issues detract from the value and contribution of this research. A central motivation for the study, stated in the abstract and again on p. 2, appears to be that a “lack of socioeconomic diversity is likely to deeply shape the type of scholarship and scholars that faculty produce and train.” However, only two citations are given in support of this claim, a study of “combating anti-blackness in the ai community” and a book that does not appear to be focused on faculty, but instead more generally on “the power of diversity” in “groups, firms, schools, and societies.”

We agree that this motivation of our study deserves more development. As another reviewer pointed out in their comments, the relationship between the sociodemographic diversity of the professoriate and the scholarship they produce is not a well-researched subject, even though there is ample related evidence that either suggests there should be a connection there, or shows evidence of such a connection in some particular context. A central goal of our study was to more clearly and quantitatively understand the socioeconomic component of current diversity

in the professoriate, which we view as an essential and valuable step in itself, which will inform and support future work aimed at understanding the relationship between diversity and scholarship.

To help make the existing evidence supporting this connection more clear to the reader, and to better articulate how our study integrates with and furthers this direction of research, we have added several more recent references that support our claim (including Kozlowski et al. *PNAS* 2022, which was published after our initial submission).

Moreover, the conclusion (p. 8) admits the lack of empirical evidence regarding the effect of background on scholarship: “..it remains unknown how socioeconomic background influences the type of scholarship that faculty pursue.” The explosion of research on race, racism, social justice, gender discrimination, and inequality, etc., across disciplines, would appear to belie the implication (not stated overtly) that the high socioeconomic background of faculty would lead to conservative scholarship, diverting attention from such topics.

We thank both this reviewer and another reviewer for raising this concern.

To better explain these ideas to the reader, in the introduction of the manuscript, we have added references and discussion of three recent studies which quantitatively examine how researcher identity shapes research topics [Kozlowski et al. “Intersectional inequalities in science” *PNAS* (2022); Hofstra et al. “The Diversity-Innovation Paradox in Science” *PNAS* (2020); Hoppe et al. “Topic choice contributes to the lower rate of NIH awards to African-American/black scientists” *Science Advances* (2019)]. In our discussion, we now also draw upon these studies to point out some of the research topics that they find are often pursued by minoritized researchers (e.g., public health and gender based discrimination).

We also draw a relationship to Bourdieu's “The specificity of the scientific field and the social conditions of the progress of reason” *Social Science Information* (1975) which explains how different scientific strategies (such as research topic choices) are related to amounts of social capital (referring to the differing resources of lower SES or POC faculty) and the investment / disinvestment of that capital.

The point we were trying to convey is that despite understanding that social capital relates to research choices, it is difficult to actually identify quantitatively the set of unstudied research topics or to model how changes in representation might change the kind of discoveries or contributions that are made. These are hard but important counter-factual questions, and at this

point, we as a field do not fully understand what we (as a community) are missing out on. By providing clear and quantitative assessments of current diversity, and its connection to socioeconomic background, our study represents a critical step toward achieving that larger goal.

We fully agree that these are important points to convey clearly to the reader. In the revised version, we have endeavored to better highlight that this is what we meant by “unknown.” We believe that addressing these points has greatly improved our paper’s framing.

Almost 1 of 5 references date from the 1990’s or earlier. Surely more recent work documents intergenerational transmission and the difficulties of those from working class backgrounds in higher educational and professional contexts. The authors may find Paying for the Party (Hamilton and Armstrong 2013) instructive.

We thank the reviewer for this suggestion. In the revised version, we now cite a number of more recent studies on various aspects of the relationship between family socioeconomic background and higher education. The updated references (including Armstrong and Hamilton 2015) illustrate the richness and continuity of scholarship in this area.

Not surprisingly, given the well-established linkages between family background and educational attainment, and the near-universal requirement that faculty in Ph.D.-granting departments have Ph.D.’s, U.S. faculty are found to be many times more likely than the general population to have parents who are Ph.D. recipients. It is also to be expected that faculty with Ph.D. parents would receive more support from their parents for their academic careers, given parents’ greater familiarity with academia. These findings are not news. The findings do indicate a clear disadvantage for racial/ethnic minorities who are less likely to have parents with doctorates (Table S3).

We thank the reviewer for this summary of our work. We fully agree that a linkage between family background and educational attainment (typically measured in terms of college completion or masters/professional training) is well documented in the literature, and that from this connection, it is reasonable to expect that linkage to also exist at the level of doctoral education. Indeed, that expectation is partly what motivated our study.

Our study quantitatively measures the extent and shape of this association, and its covariance with other factors, revealing, as the reviewer highlights, the clear intergenerational disadvantage

that racial/ethnic minorities face, as a result. We also want to point out that the broad, cross-disciplinary design of our study allows us to quantify the general population effects that past literature on family background and educational attainment predict, to show that the effects vary only modestly across broad disciplinary lines, and to provide many further analyses, showing just how pervasive the pattern truly is among tenure track faculty at research intensive universities. There is no prior work in this space with the scale, breadth, or quantitative detail of our study. We anticipate that these results will substantially inform many discussions around diversity, equity, and inclusion in academia by providing a clear picture of the current, multi-dimensional association between family background and faculty attainment.

To make these motivations and the scientific utility of our findings more clear to the reader, we have adjusted the language in the Introduction and Discussion.

The comparison in Table 1, showing that 11.8 percent of Ph.D. recipients have parents with Ph.D's, compared to 22% of those with a faculty job purportedly illustrates the great advantage of having a Ph.D. level parent. However, many of those who did not enter academia (for example, in highly technical STEM fields) after receiving a Ph.D., may have been aiming for higher paying non-academic employment in industry, think tanks, etc., and not even seeking jobs in higher education.

The reviewer makes a reasonable point, and we agree that many people pursuing PhDs are likely not aiming for faculty positions after graduation---a situation that has likely become more common in more recent decades, compared to the middle 20th century. We also agree that STEM fields are more likely to be places where people may seek a PhD without wanting to pursue a faculty career, given the statistically higher remuneration rates for non-academic jobs that rely on those degrees. It's interesting to note that this tendency is likely less in fields like Anthropology and History, which do not have clear non-academic career tracks (in contrast to most STEM fields), and yet, we find essentially the same general statistical pattern in those fields as in the more technical fields in our study. This commonality suggests that the explanation for the pattern we observe is unlikely to depend strongly on the "pull" of high-paying "in-field" non-academic jobs. Further investigation of the reasons for this similarity across fields with such different post-grad non-academic career paths would be a useful direction for future work, which does not detract from the utility of knowing that faculty are on average between 12 to 25 times more likely to have a parent with a PhD than the general population, and about twice as likely as other individuals who hold a PhD.

To highlight these interesting points for the reader, we have added some additional discussion in

the Conclusions.

It is noted (p. 6) that “nearly a third of faculty at top ranked universities across all eight fields report that one of their parents holds a PhD” (29.5%). Looked at another way, might one argue that the majority of even the most elite faculty (~60%) are upwardly mobile, more highly educated than their parents and without the socialization and other advantages that a parent with a Ph.D. would confer?

We agree that a reader could look at the numbers that way. Whether to interpret the fraction as ‘high’ or ‘low’ is something of a philosophical question and depends on the ‘null model’ a reader is using as their reference point. As we describe in the Introduction of the paper, viewing these numbers as indicating over-representation of higher socioeconomic status individuals is consistent with a literature suggesting a strong association between parents’ socioeconomic status and children’s status among advanced degree holders. Our findings deepen our understanding of social reproduction at the highest levels of academic attainment.

With respect to the methods, the 17% response rate is cause for concern (even though this may be typical of online surveys). It is stated (p. 2) that “..respondents are generally representative of their fields by institutional prestige and faculty rank,” and Table S1 shows few differences. But given the very low response rate, more attention should be given to this issue. The extent to which those who chose to respond to the survey were different in terms of socioeconomic origins from other faculty who did not respond is not discussed.

We thank the reviewer for raising this point. Of course, socioeconomic origins of those who did not respond to our survey is unknowable. However, to further elaborate on these survey methodology points, we have added some discussion around where we introduce the survey in the Data & Methods section. Moreover, in response to another reviewer, we have included the survey response rate in the Introduction, to better contextualize our results.

Given considerable evidence that college degrees are more important in the labor force than years of college attendance (see work by James Rosenbaum), it is unclear why the survey measure of parental education distinguished less than four years and four years of college education. Instead, it would be more illuminating to inquire about college degree receipt (BA or BS) and some college without degree.

The reviewer raises a reasonable point. Our survey did not ask respondents for finer grained information on the type of degree (BA or BS) their parents' achieved. For parents of our survey respondents, "four years" would typically indicate a complete college degree, and survey respondents are likely to interpret it as such, even if the degree actually took longer to complete, e.g., because "4-year degree" is the popular term and children are more likely to know whether their parent attended but did not complete college or completed college rather than the precise number of years they were enrolled. We also note that our measure aligns with a substantial amount of literature in social stratification and mobility estimating parental education via years of schooling (e.g., the National Longitudinal Surveys of the Bureau of Labor Statistics).

While it would have been ideal to gather both years of schooling and degree completion, the question's wording was chosen so as not to extend the length of the already long survey. While we wish our results could speak to the portion of the literature the reviewer highlights, unfortunately, we cannot, with this study.

Also, the measure of race departs from the current standard. Since persons of Hispanic ethnicity can be any race, the US Census and most surveys now ask first, Are you Hispanic or Latino (yes-no) followed by What is your race (White, Black or African American, etc.).

We thank the reviewer for raising this point, and we acknowledge that our survey question followed the 2010 Census classification standards for race and ethnicity. We also note that current social survey methods have not yet standardized on the new Census classification, in part because such a substantial portion of scholarship on social inequalities is built on the old standard. Our use of the old standard does allow our results to be more directly comparable with past literature on socioeconomic background. In practice, studies show there is substantial consistency across old and new standards of measurement, but we acknowledge that the new classification would enable other, more fine-grained analyses, and it would be inline with future work in this area.

To raise these contextual points for the reader, we have elaborated on this limitation of our survey design in the Conclusion, as well as highlighted the consistency between new and old measurements.

It is unclear why the coefficients for parents' highest degree in Model III, Table II, are all negative. Shouldn't those for college, masters and Ph.D. lead to higher institutional

prestige than some college?

We thank the reviewer for flagging this point. Because institutional prestige is a ranking, the most prestigious schools have rank 1 and the lowest have a rank of 100. Hence, a negative coefficient implies placement at more prestigious schools (those with rank closer to 1).

We have edited the text around Table II to clarify this point for other readers

The authors are aware that the aggregate income data is problematic, given the heterogeneity of ZIP codes. However, it is to be expected, given the education of parents, that parental incomes of faculty members would be greater than the general population.

We agree that, given the higher educational attainment of faculty parents that we find, we should expect those households to have higher than average incomes. However, as with other parts of our analysis, knowing the precise relationship by which those incomes are higher is useful information, as it speaks to the strength of the association, rather than simply whether or not it exists. In our analysis, the results around aggregate household income can be seen as supporting evidence of the broader narrative about the importance of parental education in this career.

To clarify that these results support and align with our other results, and provide a useful quantification of an association that we might expect, we have added some extra text around this section for the reader.

In sum, in view of these problems, I do not believe that the findings presented here represent a significant enough advance in knowledge to be of interest to scholars in my field or in other academic field.

We believe that in addressing reviewers' comments our paper has improved greatly, and we think the new narrative more appropriately situated these findings. We thank the reviewer for pushing us in these directions.

Response to Review 3

Reviewer #3: sociology of education, inequality in higher education

Thanks for the opportunity to read this interesting and valuable analysis. Overall this

paper makes important contribution to broader examinations of inequality in academia by focusing on faculty members' socioeconomic and intersectional backgrounds, which have been comparatively neglected in previous research. Overall I have few suggestions or critiques to make, which I hope will be helpful to the authors as they revise.

We thank this reviewer for their positive assessment of our work, and for their helpful comments. Addressing these points have improved our manuscript.

Framing

The authors note in the abstract that “Tenure-track faculty play a special role in society...” However, it’s not only TT faculty who make these contributions, but also lecturers, contingents/adjuncts, etc. TT faculty have economic and employment privileges and these jobs are much harder to get. I think this distinction is important to clarify in the framing. I agree with the focus on TT, but think there’s more to be parsed out here.

We agree with this point. Tenure track faculty represent just one slice of this ecosystem, and that many other people contribute to scholarship and the training of future scholars. We agree that the relative financial and job security of TT faculty positions, especially compared with adjunct roles, strongly enhances the ability of TT faculty to make such contributions.

To make these points more clear to the reader, we have added this context to our introduction, where we introduce the population of study, and in our discussion of our survey frame in Data & Methods.

Similarly, the article focuses only on those who have obtained TT jobs, leaving aside adjuncts or other contingent faculty and those who left academia, which should be acknowledged. Again, I don’t see this sampling choice as unreasonable (especially given the comparisons drawn among PhD holders) but think the authors are missing some important contextual dynamics or forms of inequality that shape these outcomes.

Yes, this is a fair point. We fully agree that there is some evidence to suggest that SES variables may correlate in different ways with who holds non-TT vs. TT faculty positions, and who leaves academia (at different points in the academic pipeline, e.g., before graduate school, after graduate school, after a postdoc, etc).

To make these related ideas more clear to the reader, we have adjusted the text in the introduction

and discussion to better highlight these related questions, and hence to better focus the reader's attention on the TT faculty group we consider.

Finally, it seems like there should be some discussion or at least note of racism, classism, and gender (and the intersections thereof) facing graduate students and faculty members that may drive down both hiring rates and intention to persist, as well as placements and tenure among other achievements in ways that, as the authors note, are also related to SES.

We fully agree that SES may play a causal role in many aspects of an academic career, and existing literature suggests that it likely also intersects with other forms of discrimination at different stages of an academic career. In the revised manuscript, we expand our discussion of racism, classism, and sexism to better highlight these connections and directions for future research.

Clarifying wording: The authors' wording is unclear in some places.

P1: "Working class or poor parents are less likely to enroll their children in organized activities or allow their children to question establishments..." – This is true, but it should be clear that this is at least in part structural—less access, fewer resources.

We agree. Building from other reviewer comments on our introduction, we have rewritten our introduction to focus on the educational stratification literature. We eliminated that sentence from the text.

Additionally, we have revised the manuscript to avoid individualistic explanations for these results.

P1: This sentence is a little confusing—"The association between the economic circumstances of parents and their children is often less strong (upward social mobility) for children who complete college than for those who do not [17, 18]." – does this include continuing generation children?

We have eliminated this sentence in the revised manuscript, and rewritten the paragraph to more clearly describe the educational stratification literature.

P1: The authors write that, “Occupations specifically play an important role in the intergenerational transmission of class [25], and an association between children and their parents’ occupations (sometimes called microclass reproduction or inequality) has persisted for decades [26]. This pattern is not itself problematic, but can be symptomatic

of social constraints on personal choice that systematically limit social mobility [27]” – Given the discussion so far, I’m not sure why is it not problematic.

We agree. In the revised version, we eliminated this sentence and have rewritten this paragraph.

P2: “Nevertheless, it is known that faculty are increasingly representative of highly educated families [28]...” – increasingly over time? That would surprise me. This citation is also a bit dated. I don’t think this particular point is needed, given that this is a big part of what the article seeks to establish in the authors’ own findings.

Ref. 28 is a book from 1998 that compared demographic patterns of academic faculty, finding evidence that faculty are increasingly from highly educated families. But, we agree with the reviewer that this point is unnecessary for our introduction, and also reflects a fairly narrow and more dated study. In the revision, we have removed this line.

Other minor issues

Is it the case that children of lower-income families are less likely to complete elementary school in the US (as the authors claim on P1)? The citation given is cross-national, which seems either not relevant or unclear—the article appears to be framed around US SES dynamics, not international. If that’s not the case, the authors should indicate that they include faculty respondents from cross-national backgrounds. 1999 also seems like a somewhat distant point in the past for this kind of citation.

We agree that Ref. 11 is dated. We have heavily revised the introduction of this work to be more relevant and clear and eliminated reference to elementary education.

There are faculty in our sample which may have been born outside of the U.S., but our survey design and the coverage of responses to the relevant survey questions do not allow us to precisely estimate that number. Other estimates in the literature suggest that the rate of foreign-born faculty is modest across the fields we study [e.g., see Kim, Wolf-Wendel, & Twombly, “International Faculty: Experiences of Academic Life and Productivity in U.S. Universities” *The Journal of Higher Education* (2011)]. This is all to say that while our study population is well-delineated around U.S.-based tenure-track faculty, that frame is mildly cross-national in terms of place of origin. In that sense, Ref 11 is reasonable in that it too presents a cross-national analysis.

To make these subtleties more clear to the reader, we have added text to clarify this point in our discussion of the survey frame early in the Data & Methods section.

In the discussion of Fig. 6, what accounts for the much higher and much more volatile rates of PhD attainment among Asian adults?

This is an interesting question, but also one that we feel is outside the scope of our study. These particular data are drawn from the US Census Bureau and plotted here mainly to provide context to our other results. There are numerous reasons this trend based on data from the ACS could change over time, such as due to variations in the sampling methodology over time, or immigration, or population growth. Generally speaking, we believe this variation does not reflect meaningful changes in PhD completion so we have applied an average trendline, binning every three years.

The authors make strong points about how race/ethnicity and SES, in particular, are related in these patterns. This seems worth reiterating at the start of the conclusion where the main findings are recapped.

We thank the reviewer for their suggestion. In the revision, we have adjusted the text accordingly.

Decision Letter, first revision:

Our ref: NATHUMBEHAV-210715943A

11th May 2022

Dear Dr. Morgan,

Thank you for submitting your revised manuscript "Socioeconomic Roots of Academic Faculty" (NATHUMBEHAV-210715943A). It has now been seen by the original referees and their comments are below. As you can see, the reviewers find that the paper has improved in revision. We will therefore be happy in principle to publish it in Nature Human Behaviour, pending minor revisions to satisfy the referees' final requests and to comply with our editorial and formatting guidelines.

We are now performing detailed checks on your paper and will send you a checklist detailing our editorial and formatting requirements within a week. Please do not upload the final materials and make any revisions until you receive this additional information from us.

Sincerely,

Arunas Radzvilavicius, PhD
Editor
Nature Human Behaviour

Reviewer #1 (Remarks to the Author):

I am impressed with the thorough revision of the paper and the careful attention to reviewer comments and suggestions. I especially appreciate the careful revision of the framing of the paper (and the openness of the authors to adding an author with relevant expertise).

The references still need a bit of cleanup. There is an error reference 9. The periods are inconsistently located at the end of the references. Armstrong & Hamilton was originally published in 2013 (with the paperback in 2015).

Reviewer #3 (Remarks to the Author):

The revisions completed have strengthened this work substantially. The authors have largely addressed the points I raised in my earlier comments. I have two small suggestions stemming from these revisions, however. The first, more substantial, suggestion concerns the framing. Although compelling, the authors' current wording suggests that one finding will be what "how representational diversity influences which and what kind of discoveries are made" (p. 3). Although the authors address this issue in their conclusion, it is inferred through discussion of existing work, i.e. it is not a finding--this manuscript does not directly address this issue. I agree that this is a likely and meaningful outcome and a reasonably motivation, but suggest a tweak to wording here. Second, in the authors' concluding statements (p. 10) regarding the intersection of race and SES disadvantages, I suggest adding the word "also" -- i.e. "Considering the intersection between race and class, we *also* conclude that this dependence on parent educational education and SES likely limits racial diversity within the professoriate." SES remains the primary focus of this examination, to my reading, and/but the intersection with race--and the implications for racial diversity--is also important. Finally, a copy-edit is needed for reference 9.

Author Rebuttal, first revision:

Response to Review 1

I am impressed with the thorough revision of the paper and the careful attention to reviewer comments and suggestions. I especially appreciate the careful revision of the framing of the paper (and the openness of the authors to adding an author with relevant expertise).

We thank this reviewer for their time and thoughtful comments. We are pleased to hear they are pleased with our revision.

The references still need a bit of cleanup. There is an error reference 9. The periods are inconsistently located at the end of the references. Armstrong & Hamilton was originally published in 2013 (with the paperback in 2015).

We appreciate both reviewers highlighting this typo. We have cleaned up this reference and added the correct publication date.

Response to Review 3

The revisions completed have strengthened this work substantially. The authors have largely addressed the points I raised in my earlier comments. I have two small suggestions stemming from these revisions, however.

We are glad to hear this reviewer approves of the changes, and also appreciate this reviewer's very helpful comments on the original submission.

The first, more substantial, suggestion concerns the framing. Although compelling, the authors' current wording suggests that one finding will be what "how representational diversity influences which and what kind of discoveries are made" (p. 3). Although the authors address this issue in their conclusion, it is inferred through discussion of existing work, i.e. it is not a finding--this manuscript does not directly address this issue. I agree that this is a likely and meaningful outcome and a reasonable motivation, but suggest a tweak to wording here.

We appreciate this comment. We agree this sentence should have been worded slightly more carefully. We have made a minor change here to highlight that this result is not directly implied by our work, but supports other work studying the link between representational diversity and scientific knowledge.

Second, in the authors' concluding statements (p. 10) regarding the intersection of race and SES disadvantages, I suggest adding the word "also" -- i.e. "Considering the intersection between race and class, we *also* conclude that this dependence on parent educational education and SES likely limits racial diversity within the professoriate." SES remains the primary focus of this examination, to my reading, and/but the intersection

with race--and the implications for racial diversity--is also important.

We thank the author for this suggestion. We have added the word also.

Finally, a copy-edit is needed for reference 9.

We appreciate both reviewers highlighting this typo. We have cleaned up this reference and added the correct publication date.

Final Decision Letter:

Dear Dr Morgan,

Thank you once again for your work on the requested edits. We are pleased to inform you that your Article "Socioeconomic Roots of Academic Faculty", has now been accepted for publication in *Nature Human Behaviour*.

Please note that *Nature Human Behaviour* is a Transformative Journal (TJ). Authors whose manuscript was submitted on or after January 1st, 2021, may publish their research with us through the traditional subscription access route or make their paper immediately open access through payment of an article-processing charge (APC). Authors will not be required to make a final decision about access to their article until it has been accepted. IMPORTANT NOTE: Articles submitted before January 1st, 2021, are not eligible for Open Access publication. Find out more about Transformative Journals

With best regards,

Arunas Radzvilavicius, PhD
Editor
Nature Human Behaviour